# FLEXICODEC: A DYNAMIC NEURAL AUDIO CODEC FOR LOW FRAME RATES

**Jiaqi Li**[1]  **Yao Qian**[2]  **Yuxuan Hu**[2]  **Leying Zhang**[3]  **Xiaofei Wang**[2]  **Heng Lu**[2]
**Manthan Thakker**[2]  **Jinyu Li**[2]  **Sheng Zhao**[2]  **Zhizheng Wu**[1,4,5,6]
[1]The Chinese University of Hong Kong, Shenzhen, [2]Microsoft, USA, [3]Shanghai Jiao Tong University, [4]Shenzhen Loop Area Institute, [5]City University of Macau, [6]Amphion Technology Co., Ltd.

## ABSTRACT

Neural audio codecs are foundational to speech language models. It is expected to have a low frame rate and decoupled semantic and acoustic information. A lower frame rate codec can reduce the computational cost of speech language models by shortening the sequence length. Recent studies have developed 12.5Hz low-frame-rate audio codecs, but even lower frame rate codecs remain under-explored. We find that pushing existing audio codecs to very low frame rates loses much semantic information. We suggest that low-frame-rate codecs' limitations are in both insufficient semantic decoupling and insufficient time resolution at capturing transient phonetic details. This paper introduces **FlexiCodec** to address these limitations. FlexiCodec improves semantic preservation with a **dynamic frame rate** approach and introduces a novel architecture featuring an **ASR feature-assisted dual stream** encoding and Transformer bottlenecks. With dynamic frame rates, it uses less frames at information-sparse regions through adaptively merging semantically similar frames. A dynamic frame rate also allows FlexiCodec to support inference-time **controllable frame rates** between 3Hz and 12.5Hz. Experiments on **6.25Hz, 8.3Hz and 12.5Hz** average frame rates confirm that FlexiCodec excels over baseline systems in semantic information preservation and delivers a high audio reconstruction quality. We also validate the effectiveness of FlexiCodec in language model-based TTS. Audio demos are available at: `https://flexicodec.github.io`. Code is available at: `https://github.com/amphionteam/flexicodec`.

## 1 INTRODUCTION

The neural audio codec technique, originally designed for waveform compression (Zeghidour et al., 2021; Défossez et al., 2022), is now widely used in various tasks including Language Model (LM)-based text-to-speech (TTS) (Wang et al., 2023; Anastassiou et al., 2024; Du et al., 2024a; Wang et al., 2025b; Guo et al., 2024) and multimodal LLMs (Défossez et al., 2024; Zeng et al., 2024a; Ding et al., 2025). By compressing raw speech into compact discrete tokens, neural audio codecs enable the application of auto-regressive LLM paradigms to the speech domain.

The standard neural audio codec follows an encoder-quantizer-decoder architecture. The encoder downsamples the audio waveform into a sequence of fixed-rate continuous latent vectors. These vectors are then quantized into discrete indices using Residual Vector Quantization (RVQ), a multi-stage process where each quantizer encodes the residual error of the previous one. In downstream tasks like TTS, the first RVQ layer's tokens (RVQ-1) are often used to drive an autoregressive (AR) language model or an LLM, while the remaining layers (RVQ-rest) can be predicted by a non-autoregressive (NAR) model to add acoustic detail.

However, a fundamental challenge arises from the high temporal resolution of these codecs. State-of-the-art models like EnCodec (Défossez et al., 2022), DAC (Kumar et al., 2024), and Speech-Tokenizer (Zhang et al., 2023a) operate at frame rates exceeding 50Hz, representing one second of audio with over 50 tokens. This high density, compared to the $\sim$4.5Hz rate of typical text representations (Wang et al., 2024), creates two major problems for AR models: (1) a significant computational

Table 1: Comparison of different audio tokenization methods and their properties.

| Method | Frame Rate (Hz) | Dynamic Rate | Controllable Rate | Semantic Augmentation | TTS Oriented? |
|---|---|---|---|---|---|
| DAC | 75 | ✗ | ✗ | ✗ | ✗ |
| SpeechTokenizer | 50 | ✗ | ✗ | SSL feature (HuBERT) | ✓ |
| CodecSlime | 40 (Avg) | ✓ | ✓ (40,50,67,80Hz options) | ✗ | ✗ |
| Mimi | 12.5 | ✗ | ✗ | SSL feature (WavLM) | ✓ |
| DualCodec | 12.5 / 25 | ✗ | ✗ | SSL feature (w2v-bert-2) | ✓ |
| TaDiCodec | 6.25 | ✗ | ✗ | Text | ✓ |
| Phoneme Tokens | 11.7 (Avg) | ✓ | ✗ | - | - |
| BPE Text Tokens | 4.5 (Avg) | ✓ | ✗ | - | - |
| **FlexiCodec** | **6.25 / 8.3 / 12.5 (Avg)** | ✓ | **✓ (Any from 3 to 12.5Hz)** | **ASR feature** | ✓ |

burden due to the quadratic complexity of attention, and (2) a severe frame rate mismatch between text and audio modalities that may degrade LLM performance (Zeng et al., 2024b; Wang et al., 2024).

To mitigate this, recent work has focused on low-frame-rate codecs. Methods like Mimi (Défossez et al., 2024) and DualCodec (Li et al., 2025a) successfully reduced the frame rate to 12.5Hz by decoupling speech into two streams: a semantic stream (RVQ-1) derived from self-supervised learning (SSL) models (Chen et al., 2022; Barrault et al., 2023), and an acoustic stream (RVQ-rest) for residual details. In this strategy, the low-rate RVQ-1 tokens encode the core semantic information, which is sufficient for many downstream AR models.

While previous works have proposed 12.5Hz solutions, a significant gap remains compared to the ~4.5Hz frame rate of text. Furthermore, research into neural audio codecs operating below 12.5Hz is limited. Our initial experiments revealed that pushing existing codecs to below 12.5Hz leads to significant loss of semantic/phonetic content. We suggest two reasons for this result. First, at very low frame rates, the limited information capacity forces a trade-off between acoustic fidelity and semantic representation. Existing codecs may be primarily limited by an insufficient decoupling of semantics from acoustics. This is exemplified by recent work on syllable-level unit discovery (Cho et al., 2024a; Baade et al., 2024), which successfully extracts semantic units at 5-8Hz but largely discards the fine-grained acoustic details, prosody, and timing required for high-fidelity reconstruction. We discuss these works in Section 2.2. Second, we propose that current fixed-rate downsampling can lead to the loss of transient phonetic details, whereas natural speech units like syllables and phonemes are inherently dynamic in their rate of occurrence. Driven by these observations, we have the following motivations for designing our **low frame rate** codec: (1) **Dynamic Frame Rate**: A fixed low rate inevitably discards transient phonetic details, whereas a dynamic rate may adapt to the phonetic complexity to encode more details. (2) **Richer Semantics**: SSL features, trained for reconstruction, can be redundant. Features from an ASR model, trained for text prediction, may offer a more concentrated source of semantic information. (3) **Controllable Frame Rate**: Existing codecs typically operate at one or more fixed frame rates. A continuously controllable rate would allow users to dynamically trade off performance and efficiency for downstream tasks.

In this work, we introduce FlexiCodec, a novel low-frame-rate codec built on three principles: dynamic frame rate, ASR-guided semantics, and frame rate controllability. Instead of a fixed frame rate, FlexiCodec dynamically allocates temporal resolution, using more frames for complex phonetic segments and fewer for sparse regions like long vowels, syllables and silence. This is achieved through a novel ASR-feature-assisted dual-stream architecture that adaptively merges semantically similar frames. A key benefit of this dynamic approach is that a single model can support a continuous range of frame rates **(3-12.5Hz)** at inference, enabling flexible trade-offs for applications like adaptive signal transmission or variable-complexity TTS on edge devices. Our contributions are:

- We propose FlexiCodec, a novel codec that pushes the boundaries of low-frame-rate audio tokenization. To our knowledge, it is among the first codecs to achieve high-quality, reconstructible audio at average frame rates **below 10Hz**. We also introduce a novel method for **dynamic frame rate allocation in the low-frame-rate regime (3-12.5Hz)**, guided by ASR features. A review of recent below 10Hz codecs and dynamic-rate codecs are provided in Sections 2.1 and 2.2.

- We show that FlexiCodec outperforms open-source baselines in semantic intelligibility and acoustic quality. Experiments confirm that our dynamic frame rate strategy improves semantic information preservation and allows for controllable frame rate as low as 3Hz. Other

design choices including utilizing an ASR encoder, transformer bottlenecks, and FSQ quantization, also contributes to our codec's performance.

- We demonstrate FlexiCodec 's utility in a flexible TTS system. The model yields competitive results at multiple frame rates and is substantially faster than existing methods.

## 2 RELATED WORKS

### 2.1 LOW-FRAME-RATE NEURAL AUDIO CODECS

Neural audio codecs convert continuous speech into discrete tokens. SoundStream (Zeghidour et al., 2021), Encodec (Défossez et al., 2022), and DAC (Kumar et al., 2024) focused on audio compression, relying on residual vector quantization (RVQ) (Zeghidour et al., 2021) and operating at high bitrates ($\geq$4kbps) and high frame rates ($\geq$50Hz). WavTokenizer (Ji et al., 2024), TS3-Codec (Wu et al., 2024), SemantiCodec (Liu et al., 2024), and StableCodec (Parker et al., 2024) used a single VQ or FSQ (Mentzer et al., 2023) codebook. They delivered good audio quality at low bitrates (around 1kbps), but operate at a high frame rates ($\geq$40Hz).

Some recent works develop low-frame-rate codecs. A lower frame rate limits the information amount that can be carried by each RVQ layer tokens. Thus, some works (Défossez et al., 2024; Li et al., 2025a) decompose the speech tokens into semantic (RVQ-1) and acoustic (RVQ-rest) tokens. Mimi codec (Défossez et al., 2024) was based on Encodec with a higher downsampling rate in its convolutional encoder. It employed semantic distillation (Zhang et al., 2023a), a technique that distills RVQ-1 embeddings from SSL features. DualCodec (Li et al., 2025a) proposed a dual-stream architecture where a semantic stream directly encodes SSL features into RVQ-1 tokens, and an acoustic stream encodes RVQ-rest tokens. ALMTokenizer (Yang et al., 2025b) proposed a query-based compression strategy using a set of learnable query tokens, and designed an auxiliary MAE loss inspired by SSL models. Concurrent work XY-Tokenizer (Gong et al., 2025) is a 12.5Hz codec encoded with a concatenative dual-stream architecture consisting of Whisper ASR feature and waveform feature.

Recently, TaDiCodec (Wang et al., 2025c) and TASTE (Tseng et al., 2025) proposed $\leq$10Hz speech tokens, but operating like text-to-speech (TTS) systems, they require the text transcription to assist audio synthesis. Accurate text transcription can be unavailable in some audio coding scenarios. By comparison, our work is more similar to a conventional codec and does not require transcriptions.

### 2.2 DYNAMIC-RATE COMPRESSION OF IMAGES AND AUDIOS

The concept of dynamically adjusting token rates based on content complexity is an emerging trend across image and audio modalities. In the image domain, Bolya et al. (2022) first proposed Token Merging (ToMe), a technique to gradually merge most similar tokens in Vision Transformers (ViTs) to accelerate inference. It used the cosine similarity between the self attention keys to guide the merging process. DynTok (Zhang et al., 2025b) proposed an improved similarity calculation strategy based on CLIP (Radford et al., 2021) semantic representation.

In the audio domain, one research trend has been the syllable-level semantic unit discoveries which are inherently dynamic-rate. SD-HuBERT (Cho et al., 2024b) finetuned HuBERT (Hsu et al., 2021) with sentence-level self-distillation, and showed that its features distinguish syllable boundaries. Sylber (Cho et al., 2024a) leveraged SD-HuBERT as pseudo labels, and trained a student model with syllable encoding ability at around 5 syllables per second. SylBoost (Baade et al., 2024) can discover discrete syllabic units using a min-cut algorithm on the feature self-similarity matrix, followed by a k-means clustering. It experimented on 8.33Hz and 6.25Hz units. However, compared to neural audio codec tokens, syllable units only encode coarse semantic information, and necessitate external speech synthesis models to produce audio.

Dynamic-rate neural audio codec is an emerging research area. Dieleman et al. (2021) proposed an audio VQ-VAE with run-length encoding, operating at an average frame rate of 75Hz. SNAC (Siuzdak et al., 2024) created a multi-resolution codec stream at 12, 23, and 47Hz for each RVQ layer. CodecSlime (Wang et al., 2025a) proposed a two-stage process to firstly train an 80Hz fixed-frame-rate codec, followed by merging similar features into 40Hz average-frame-rate tokens. Similarly,

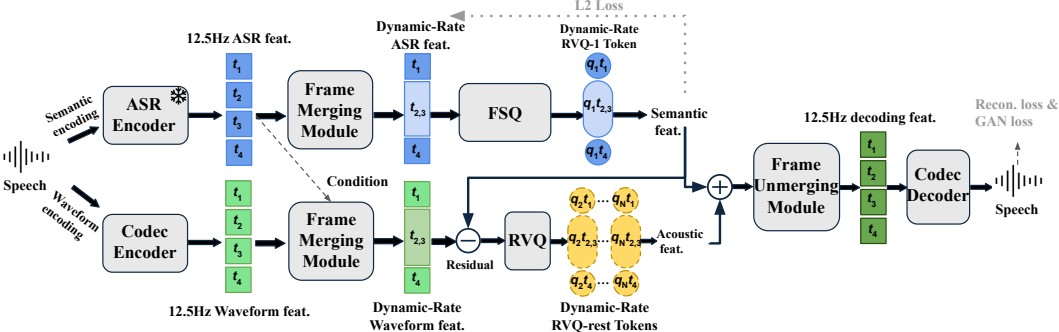

Figure 1: Overview of FlexiCodec. The model encodes speech through two streams. The Frame Merging Modules dynamically reduce the 12.5Hz features into lower frame rates, and the Frame Unmerging Module restores a 12.5Hz fixed frame rate. The model is trained end to end.

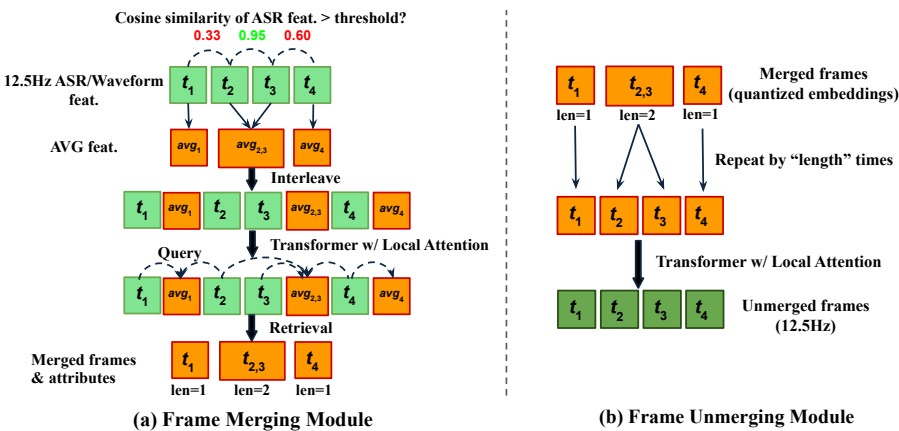

Figure 2: Detailed views of the Frame Merging Module and the Frame Unmerging Module.

TFC (Zhang et al., 2025a) and VARSTok (Zheng et al., 2025) introduced dynamic-frame-rate algorithms on 75Hz RVQ or single-codebook tokens. Compared with previous works, ours explores more challenging low frame rates ($\leq$10Hz). Compared with CodecSlime, TFC and VARSTok, our work uses a pre-trained audio semantic feature extractor to guide the merging process, simplifying the needs for multi-stage training or dynamic programming-based merging algorithms.

## 2.3 TEXT-TO-SPEECH SYNTHESIS

Modern Text-to-Speech (TTS) has increasingly shifted from statistical parametric methods to systems based on neural audio codecs. VALL-E (Wang et al., 2023) established an "AR+NAR" framework: an AR language model generates codec RVQ-1 tokens conditioned on text; an NAR language model predicts remaining RVQ layer tokens in parallel. The AR stage provides essential information for the NAR stage, and the AR stage is usually the most time-consuming part. Subsequent works have evolved this paradigm, showing the benefits of semantic-rich RVQ-1 tokens in the AR stage (Borsos et al., 2023a; Du et al., 2024a; Zhang et al., 2025d), and replacements of the NAR discrete token prediction with diffusion-based models (Betker, 2023; Du et al., 2024a).

The efficiency of a TTS system directly correlates to codec frame rate. A lower frame rate codec can significantly boost training and inference speed. A few works (Du et al., 2024b; Deng et al., 2025) have adopted 25Hz frame rate tokens in the AR stage, others mostly rely on $\geq$50Hz tokens (Guo et al., 2024; Wang et al., 2025b). For the NAR stage, these works mostly keep a frame rate $\geq$50Hz.

## 3 FLEXICODEC: A DYNAMIC LOW-FRAME-RATE NEURAL AUDIO CODEC

The core of our work is **FlexiCodec**, a neural audio codec designed to operate at very low average frame rates while preserving crucial semantic information. Unlike traditional codecs that use a fixed frame rate, FlexiCodec employs a **dynamic frame rate** mechanism that allocates more temporal resolution to information-dense regions of speech and less to information-sparse segments like silence or long vowels. This is achieved through an architecture that leverages **pre-trained ASR features to encode semantic-rich RVQ-1 tokens**, and to **guide the adaptive frame merging** process. The overall architecture, depicted in Figure 1, follows a dual-stream encoding, dynamic frame merging, quantization, and frame unmerging decoding pipeline.

**Dual-Stream Feature Extraction**    FlexiCodec begins by processing a 16kHz speech waveform through two parallel encoders to decouple semantic and acoustic information: (1) An ASR Encoder, leveraging the pre-trained ASR model, extracts a sequence of semantic features $e_s \in R^{T \times d}$, where $T$ and $d$ denote the number of frames and the vector dimension. And (2) a convolutional codec encoder, downsampling the waveform to produce a sequence of waveform features $e_a \in R^{T \times d}$, also at a 12.5Hz frame rate. The codec encoder consists of 5 CNN blocks with strides [4,4,5,8,2] to gradually downsample the audio, giving $16000\text{Hz} \div (4 \times 4 \times 5 \times 8 \times 2) = 12.5\text{Hz}$. Each CNN block contains a strided 1D convolution followed by a ResNet (He et al., 2016).

For the specific ASR model choice, we use pretrained SenseVoice-Small (An et al., 2024), a 230M-parameter, encoder-only Transformer model trained on 300k hours of data with CTC loss. We use its last layer (excluding its CTC logits prediction layers) hidden state as the semantic feature. The model outputs 16.67Hz features; we downsample it to 12.5Hz using linear interpolation to align the frame rate with the acoustic stream. The ASR model is frozen during codec training.

**Dynamic Frame Merging**    This module compresses the number of frames in a sequence, resulting in a sequence where some segments have lower than 12.5Hz frame rate. Our goal is to compress the frames with a minimal loss of semantic information by preferentially merging frames that are semantically similar (redundant). Inspired by DynTok (Zhang et al., 2025b) that used pre-trained image features for token compression, we reuse the extracted ASR feature to guide our merging process, shown in Figure 2(a). Specifically, let the 12.5Hz ASR feature vectors be $\{e_s[1], e_s[2], \ldots, e_s[T]\}$. We first compute the cosine similarity between adjacent frames: $s_t = \cos(e_s[t], e_s[t+1])$    for $t = 1, \ldots, T-1$. We then scan from left to right to form maximal contiguous segments $[i, i+1, \ldots, j]$ whose adjacent similarities exceed a threshold $\tau$: $\min_{t=i}^{j-1} s_t \geq \tau$. All frames in such a segment are merged into a single frame by averaging, applied to both the semantic and acoustic streams:

$$\tilde{e}_s[k] \ = \ \frac{1}{\ell_k} \sum_{t=i}^{j} e_s[t], \quad \tilde{e}_a[k] \ = \ \frac{1}{\ell_k} \sum_{t=i}^{j} e_a[t], \quad \ell_k \ = \ j - i + 1.$$

where $\tilde{e}_s[k]$ is the $k$-th entry in the compressed frame sequence in the semantic stream, and $\tilde{e}_a[k]$ is the sequence in the acoustic stream. We also record the length of each merged frame $\ell_k$ as attributes, which are required to reconstruct the original fix-frame-rate sequence.

We find that directly using the average-reduced sequence can degrade the naturalness of the reconstructed audio, particularly causing unnatural transitions between merged frames. To address this, we adopt a transformer with local windowed attention that processes an interleaved sequence of original and averaged frames (Fig.2(a)) (Yang et al., 2025b; Li et al., 2023). This allows the merged tokens to query their adjacent context and produce refined, context-aware representations. A local attention is favored over global attention because it allows generalization to longer audios despite trained on fixed-length audio segments.

**Frame Rate Flexibility**    FlexiCodec trains with a flexible merging threshold $\tau$, sampled across a range. This enables the model to support controllable frame rates at inference by adjusting $\tau$. At $\tau = 1.0$ (no frame merging), FlexiCodec functions as a 12.5Hz fixed-frame-rate codec. At $\tau < 1.0$ (with frame merging), the average frame rate of the new sequence is lower than 12.5Hz, and is calculated as: $\frac{\text{Total number of frames after merging}}{\text{Audio duration in seconds}}$. Setting a lower $\tau$ decreases the average frame rate.

**Semantic (RVQ-1) Quantization**    The dynamic-rate ASR features are quantized using a Finite Scalar Quantizer (FSQ) (Mentzer et al., 2023) to produce discrete semantic tokens (we denote as RVQ-1 tokens). FSQ projects the input representations $e_s$ into a $D$-dimensional low-rank space, where the value of each dimension is quantized using rounding operation ROUND into $L$ levels.

The quantized low-rank vector $\bar{e}$ is subsequently projected back to the original dimension $\hat{e}$:

$$\bar{e_s} = \text{ROUND}\big(\text{Proj}_{\text{down}}(e_s)\big), \quad \hat{e_s} = \text{Proj}_{\text{up}}(\bar{e_s}).$$

Straight-through estimation (Bengio et al., 2013) is employed to estimate the gradients in the ROUND function. The semantic token index $q_s$ is obtained by: $q_s = \sum_{j=0}^{D-1} \bar{e_s} L^j$. Additionally, we wrap the FSQ block with small ConvNeXt (Liu et al., 2022) blocks to increase its representation ability, and apply an L2 loss $L_{\text{feat}}$ to align the semantic token embeddings with the unquantized semantic features.

**Acoustic (RVQ-rest) Quantization**   To encode the detailed acoustic information that is not captured in the semantic tokens, following Li et al. (2025a), we compute a residual by subtracting the dynamic-rate ASR feature from the dynamic-rate waveform feature. This residual is then quantized using an $(N-1)$-layer Residual Vector Quantization (RVQ) (Zeghidour et al., 2021)[1] module to produce acoustic tokens $q_{2:N} \in Z^{(N-1) \times \hat{T}}$, where $\hat{T}$ denotes the sequence length after frame merging. We employ quantizer dropout (Zeghidour et al., 2021) during training. That is, only the RVQ-1 to RVQ-$n$ layers are subsequently decoded, where $n \in [1, N]$ is randomly chosen. When $n = 1$, only the semantic encoding stream is used. Straight-through estimation (Bengio et al., 2013) is used to backpropagate through the codebook lookup.

**Frame Unmerging and Reconstruction**   To reconstruct the audio, the decoder path should reverse the dynamic compression[2]. The embedding features from the first $n$ chosen RVQ tokens are added to form a decoding feature representation. This dynamic-rate sequence is then passed to the Frame Unmerging Module, detailed in Figure 2(b). It uses the frame length attributes to expand the sequence back to 12.5Hz. It is followed by another Transformer with local attention to smooth the transitions and refine the feature representations. The resulting feature sequence is then fed into a convolutional codec decoder which mirrors the codec encoder, synthesizing the output waveform.

FlexiCodec is trained end-to-end with a composite loss function:

$$\mathcal{L} = \mathcal{L}_{\text{recon}} + \lambda_{\text{GAN}}\mathcal{L}_{\text{GAN}} + \lambda_{\text{RVQ}}\mathcal{L}_{\text{RVQ}} + \lambda_{\text{feat}}\mathcal{L}_{\text{feat}}, \tag{1}$$

where $\mathcal{L}_{\text{recon}}$ is a multi-scale L1 mel spectrogram reconstruction loss following Kumar et al. (2024), $\mathcal{L}_{\text{GAN}}$ contains adversarial and feature matching losses for Multi-Period Discriminator (MPD) (Kong et al., 2020) and Multi-Resolution Spectrogram Discriminator (MRSD) (Kumar et al., 2024), $\mathcal{L}_{\text{RVQ}}$ involves a L1 codebook update loss and a commitment loss for RVQ, whereas the FSQ module does not require a training loss. $\mathcal{L}_{\text{feat}}$ is the L2 feature alignment loss between the RVQ-1 semantic token embeddings and the unquantized semantic features.

## 4   EXPERIMENTS

### 4.1   EXPERIMENTAL SETUP

**Codec Training Configuration**   We use the 16kHz, 54k-hour Librilight-Large (Kahn et al., 2020) dataset for training. Each model is trained for 800k steps on 8 Nvidia V100 32GB GPUs. We use a batch size of 5 samples per GPU; each sample is a 5 second audio segment. We use AdamW (Loshchilov & Hutter, 2017) optimizer with lr=$1 \times 10^{-4}$, betas=$(0.8, 0.99)$; exponential learning rate with gamma=0.999998. In each step, the merging threshold $\tau$ is randomly chosen from $0.7 \le \tau \le 1.0$. We set the maximum frame length $\ell_k$ to 8 so that each $\ell_k$ would take at most $log_2 8 = 3$ bits of storage. Each token in the local attention transformer can attend to $\ell_k = 8$ tokens left and right. The FSQ module has $D = 5$ dimensions each quantized to $L = 8$ levels, resulting in $8^5 = 32768$ codebook entries. The RVQ-rest quantization has 24 RVQ layers, 4096 codebook entries per layer, and 512-dimensional codebook entries. FlexiCodec has 216M trainable parameters, in which the two frame merging modules each is 20M, and the frame unmerging module is 100M.

---

[1]We have not applied FSQ for acoustic quantization because FSQ is a single-layer quantization. We consider the integration of more advanced multi-layer FSQ (rFSQ) schemes, e.g., (Parker et al., 2024), to be a promising direction for future improvements.

[2]This is because the NAR codec decoder can only receive fixed-frame-rate sequence. It is also possible to generate audios directly from dynamic-rate tokens using an AR model, and we leave this as future work.

**Codec Evaluation Metrics**   We evaluate on the 4 to 10 second subset of LibriSpeech-test-clean (Panayotov et al., 2015), comprising 1,088 audios. We evaluate one FlexiCodec on three frame rates: 12.5Hz (80ms hop size), 8.3Hz (120ms hop size), and 6.25Hz (160ms hop size). Since FlexiCodec has dynamic, controllable frame rate, we configure its $\tau = 1.0, 0.91, 0.867$ for 12.5Hz, 8.3Hz, and 6.25Hz average frame rate, respectively. These $\tau$ settings have been determined based on trial runs on our test set. Our evaluations consist of the following semantic and acoustic testings:

• **Semantic testing:**   To evaluate the preservation of semantic content, we transcribe the codec reconstructed audio using a Hubert-Large-LS960-ft (Hsu et al., 2021) ASR model and compute the word error rate (WER) against ground truth transcriptions. The testings include using RVQ-1 alone, and RVQ-1:8 (1 to 8 layers) tokens. Specifically, RVQ-1 tokens' semantic preservation relates to downstream model performance, where AR LMs only access RVQ-1 tokens (Li et al., 2025a; Zhang et al., 2023a). Evaluating RVQ-1:8 is a common choice of RVQ-based codecs, implying the acoustic compression quality (Défossez et al., 2024; Zhang et al., 2023a).

• **Acoustic testing:**   We evaluate the reconstructed audio from RVQ1:8 tokens against the original audio. Metrics include the Perceptual Evaluation of Speech Quality (PESQ, narrow band) (Rix et al., 2001), Mel Cepstral Distortion (MCD) (Kubichek, 1993), speaker similarity (SIM, the cosine similarity between speaker embeddings extracted from a WavLM-large-based (Chen et al., 2022) speaker verification model), and a speech perceptual quality score from a neural Mean Opinion Score (MOS) predictor UTMOS (Saeki et al., 2022).

• **Additional semantic testing with ASR probing:**   Additionally, to evaluate the semantic alignment between FlexiCodec semantic tokens and text, we employ an ASR probing task (Gong et al., 2025; Zhang et al., 2023a), adapted from XARES (Zhang et al., 2025c) benchmark. We train a downstream Qwen2.5-0.5B-based (Yang et al., 2025a) ASR model which is tasked with predicting the text transcription given FlexiCodec's RVQ-1 token embeddings. During training, only the parameters of an MLP adapter are updated. Models are trained on LibriSpeech train-clean-100 using cross entropy loss. We evaluate the ASR WER on LibriSpeech test-clean. The task's upper bound result is obtained by training with the unquantized SenseVoice-Small encoder feature. Due to the prolonged evaluation time of this task, we only conduct this task as an extended semantic testing in Section 4.3.

## 4.2   Examining the Impact of Very Low Frame Rates

We first investigate the performance of representative audio codecs at very low frame rates. Since open-source baselines operating below 12.5Hz are unavailable, we created three new baseline versions by retraining DAC (Kumar et al., 2024) and DualCodec (Li et al., 2025a) to operate at 12.5Hz, 8.3Hz, and 6.25Hz, respectively. To adapt these systems for lower frame rates, we increased their encoder downsampling rates and enlarged their codebook sizes to be consistent with FlexiCodec. Detailed configurations are provided in Appendix G.1. Our findings are as follows.

• **RVQ-1 tokens' semantic information preservation is challenging for very low frame rate codecs.**   As shown in Figure 3(a), we see that the Word Error Rate (WER), evaluated on audios reconstructed from RVQ-1 tokens, increases substantially for DAC and DualCodec when the frame rate drops from 12.5Hz to 6.25Hz. At 12.5Hz, DualCodec achieves a 5.93% WER, close to the ground truth (GT) 2.1%. However, as its frame rate drops to 6.25Hz, its gap with GT significantly increases to 31.5% vs. 2.1%, indicating its **lower resolution tokens fail to capture the whole phonetic details.** Note that DualCodec architecture already improves WER over DAC thanks to its semantic augmentation using SSL features. Figure 3(b) confirms this trend, though the performance gap is smaller when using more RVQ layers.

• **FlexiCodec excels at semantic preservation, especially at the lowest frame rates.** In contrast, FlexiCodec maintains a low WER across all rates. At the most challenging 6.25Hz average frame rate, FlexiCodec achieves 4.15% WER which is close to ground truth (2.1%) and outperforms the best baseline (31.5%). To explain FlexiCodec's superior semantic information retention, our further experiments suggest it is contributed by a combination of our proposed design choices, including the ASR-assisted encoding architecture, dynamic frame rate merging, FSQ, and Transformer modules. These modules are ablated in Appendix D. We have found that a simple switching from DualCodec's SSL into FlexiCodec ASR feature is very useful and it achieves an RVQ1 WER of 6.0% at 6.25Hz.

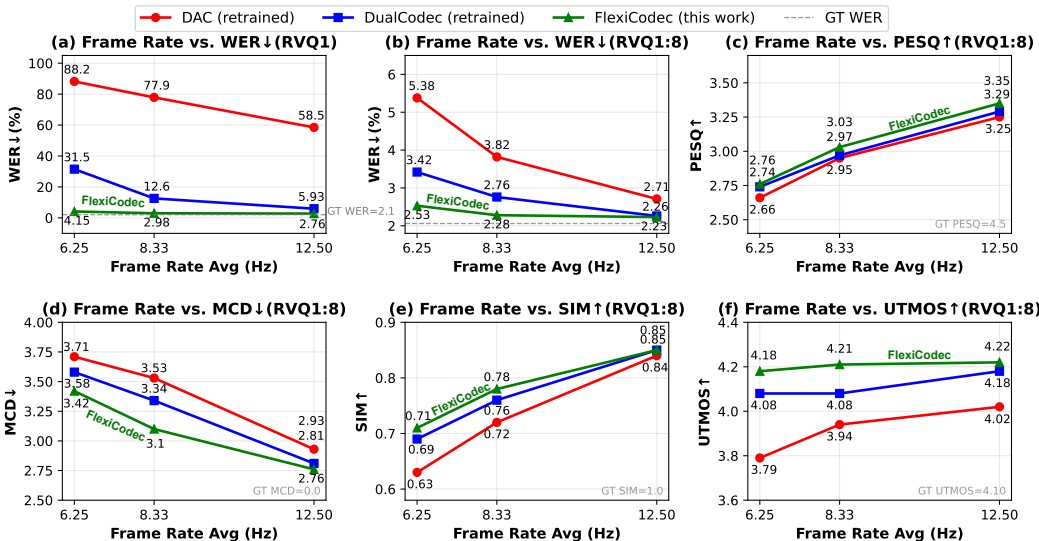

Figure 3: Evaluation results on three very low frame rates. Each baseline system has been retrained for each target frame rate using the same recipe as FlexiCodec.

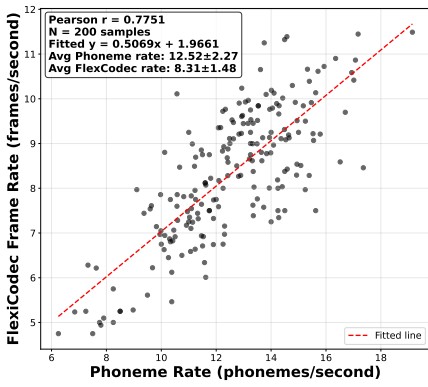

Figure 4: Correlation between Flexi-Codec frame rate and phoneme rate at a fixed frame merging threshold $\tau$. Each data point is an audio in TIMIT dataset, representing the audio's average phoneme rate vs. average FlexiCodec frame rate.

Table 2: Relation between FlexiCodec's frame merging threshold $\tau$, average frame rate, and WER.

| $\tau$ | 0.7 | 0.75 | 0.8 | 0.9 | 1.0 |
|---|---|---|---|---|---|
| AVG frame rate | 3.0 | 3.6 | 4.5 | 7.9 | 12.5 |
| WER(RVQ1)↓ | 51.5 | 29.5 | 14.4 | 3.13 | 2.76 |
| WER(RVQ1:8)↓ | 18.1 | 8.90 | 4.38 | 2.37 | 2.23 |

Table 3: Ablation of dynamic frame rate (semantic test).

|  | WER(RVQ1)↓ (diff.) | WER(RVQ1:8)↓ (diff.) | ASR probing WER↓ (diff.) |
|---|---|---|---|
| GT / Upper bound | 2.10 | 2.10 | 11.1 |
| FlexiCodec @8.3Hz | **2.98** | **2.28** | **13.0** |
| → w/o dynamic frame rate | 3.56 (+19% rel.) | 2.43 (+6% rel.) | 14.5 (+12% rel.) |
| FlexiCodec @6.25Hz | **4.15** | **2.53** | **15.6** |
| → w/o dynamic frame rate | 5.22 (+26% rel.) | 2.73 (+8% rel.) | 18.8 (+21% rel.) |

Table 4: Ablation of dynamic frame rate (acoustic test).

|  | PESQ↑ | MCD↓ | UTMOS↑ | SIM↑ |
|---|---|---|---|---|
| FlexiCodec @8.3Hz | 3.03 | **3.10** | 4.21 | **0.78** |
| → w/o dynamic frame rate | 3.03 | 3.18 | 4.21 | 0.76 |
| FlexiCodec @6.25Hz | 2.76 | **3.42** | 4.18 | **0.71** |
| → w/o dynamic frame rate | 2.76 | 3.47 | 4.18 | 0.70 |

Further improvements are particularly driven by the dynamic frame rate strategy, which we will detail in Section 4.3.

• **Acoustic quality metrics show more moderate differences across systems and frame rates.** As shown in Figure 3(c-e), the PESQ, MCD, SIM, and UTMOS metrics show that at each frame rate, FlexiCodec slightly outperforms DualCodec, followed by a larger gap with DAC. We observe that the gap between models does not substantially increase when changing to lower frame rates, but has been seen in the semantic testing. We think that the acoustic fidelity is more constrained by bitrate, and because the bitrates of the three systems are at the same level, the difference is not pronounced.

### 4.3 Analysis and Ablation of Dynamic Frame Rate

To understand the mechanism and performance of dynamic frame rate, we conduct a series of analyses and ablations shown in Figure 4 and Table 2-4. Our findings are as follows.

• **Dynamic frame merging effectively adapts to the underlying phonetic complexity of speech.**
Figure 4 shows a strong positive correlation (Pearson $r = 0.775$) between the utterance-level phoneme rate and FlexiCodec's frame rate on the TIMIT (Garofolo et al., 1993) subset. This demonstrates that FlexiCodec dynamically adjusts its token frame allocation in proportion to the phonetic density of the audio, assigning more frames to segments with faster speech and fewer frames to slower or silence regions. Such adaptivity enables efficient semantic compression by aligning token rate with semantic information density. The fitted line also shows a linear coefficient very close to 0.5, indicating that each merged frame approximately encodes two phonemes. We visualize several cases by aligning phonemes labels with FlexiCodec frames, which is shown in Appendix A. It confirms that typical merged tokens include syllables/short words, long vowels, and silence.

• **Dynamic frame rate enables controllable frame rate as low as 3Hz.** As shown in Table 2, the merging threshold $\tau$ controls frame rate and semantic preservation trade-off. Notably, by controlling $\tau$ values at inference time, we can obtain different output sequence lengths, which is a unique feature compared to conventional fixed-frame-rate codecs.

• **Dynamic frame rate improves semantic information preservation.** Tables 3 and 4 compare FlexiCodec and its variants that are retrained with fixed frame rate (FFR). To obtain the FFR variants, we modify their codec encoder strides to output static 8.3Hz or 6.25Hz, and maintain the total parameter count (detailed in Appendix G.1). Table 3 shows that the FFR variants have consistently worse WERs than FlexiCodec. The gap is larger at the lower frame rate. Specifically, the 6.25Hz FFR variant increases RVQ-1 WER by a relative 26%, the ASR probing WER by 21%, and the RVQ1:8 WER by 8%. These results highlight that dynamic frame rate improves semantic information preservation by a mechanism of phonetic complexity-adaptive frame rate allocation. We also note that dynamic frame rate may perform even better on real-world data, which tend to have longer silence regions that are highly compressible. We leave this investigation as future work.

In terms of acoustic metrics, Table 4 shows that PESQ and UTMOS are the same between FlexiCodec and its FFR variant, while a moderate degradation in MCD and SIM is observed in FFR. This indicates that dynamic frame rate mainly boosts semantic preservation rather than low-level acoustic quality. One possible reason is that the acoustic information density in a speech can be misaligned with the semantic information density. For example, semantically unimportant segments may still contain acoustic details like noise, sound and music.

### 4.4 Comparison with Open-Source Codecs at Various Bitrates

In this experiment, we compare FlexiCodec with open-source neural audio codecs spanning various bitrate [3] and frame rates. Information about the baseline codecs are provided in Appendix G. We categorize them into 3 bitrate classes, and FlexiCodec at 12.5Hz, 8.3Hz, and 6.25Hz average frame rates fall into each category, enabling bitrate-consistent comparisons. Table 5 presents the results.

• **FlexiCodec has state-of-the-art audio quality at various bitrate levels.** Examining the acoustic test scores, at >1kbps, FlexiCodec at 12.5Hz (1.2kbps) achieves higher acoustic scores than its 12.5Hz counterparts Mimi, XYTokenizer and DualCodec. It only trails behind DAC which uses a higher bitate (6kbps). At 0.8kbps and 0.6kbps, FlexiCodec also demonstrates superior acoustic quality scores than baselines. These results demonstrate that FlexiCodec has a high bitrate efficiency.

• **FlexiCodec is competitive to higher frame rate systems at semantic information preservation.**
Across the WER semantic test metrics at both RVQ1 and RVQ1:8 quantization levels, FlexiCodec consistently achieves competitive or better scores compared to other systems operating at higher frame rates. Notably, FlexiCodec at 6.25Hz attains an RVQ1 WER of 4.15, outperforming larger frame rate models such as SpeechTokenizer-50Hz, Encodec-75Hz, WavTokenizer-75Hz, etc.

---

[3] *Bitrate* indicates the amount of data stored or transmitted per unit of time, measured in kilobits per second (kbps). It can be calculated by $(\log_2 \text{Codebook\_size}) \times \text{Frame\_rate} \times \text{Num\_rvq\_layer}$. A lower bitrate codec has higher compression ratio but usually lower audio quality. Since storing and transmitting each of FlexiCodec's frame length attribute takes $\log_2 8 = 3$ bits, we have added $3 \times \text{frame\_rate}$ to FlexiCodec's bitrate in Table 5.

Table 5: Comparison between FlexiCodec and other open-source neural audio codecs.

| System | RVQ1 BR(kbps) | RVQ1:8 BR(kbps)/n_q | Param | Semantic Test | | Acoustic Test (RVQ1:8) | | | |
|---|---|---|---|---|---|---|---|---|---|
| | | | | WER(RVQ1)↓ | WER(RVQ1:8)↓ | PESQ↑ | UTMOS↑ | MCD↓ | SIM↑ |
| >1kbps Acoustic Bitrate | | | | | | | | | |
| DAC-75Hz | 0.75 | 6.0 / 8q | 74M | 31.2 | 2.27 | **3.77** | 3.62 | **2.34** | **0.90** |
| Encodec-75Hz | 1.50 | 6.0 / 8q | 15M | 5.90 | 2.24 | 3.12 | 3.01 | 2.60 | 0.89 |
| SpeechTokenizer-50Hz | 0.50 | 4.0 / 8q | 103M | 5.56 | 2.47 | 3.01 | 3.90 | 3.17 | 0.85 |
| Mimi-12.5Hz | - | 1.1 / 8q | 78M | - | 3.15 | 2.75 | 3.56 | 3.62 | 0.73 |
| XYTokenizer-12.5Hz | - | 1.0 / 8q | 520M | - | 2.36 | 3.00 | 4.00 | 3.28 | 0.84 |
| DualCodec-12.5Hz | 0.19 | 1.2 / 8q | 84M | 5.93 | 2.26 | 3.29 | 4.18 | 2.81 | 0.85 |
| FlexiCodec @12.5Hz | 0.23 | 1.3 / 8q | 216M | **2.76** | **2.23** | 3.35 | **4.22** | 2.76 | 0.85 |
| ~0.8kbps Acoustic Bitrate | | | | | | | | | |
| WavTokenizer-75Hz | 0.90 | 0.90 / 1q | 81M | 4.57 | 4.57 | 2.86 | 3.98 | 3.51 | 0.68 |
| SNAC-12,23,47Hz | - | 0.98 / 3q | 20M | - | 4.21 | 2.51 | 3.43 | 3.61 | 0.67 |
| XCodec2-50Hz | 0.80 | 0.80 / 1q | 210M | **2.80** | 2.80 | 2.77 | 4.08 | 3.65 | **0.82** |
| TS3Codec(X2)-50Hz | 0.85 | 0.85 / 1q | 204M | 4.09 | 4.09 | 2.80 | 3.80 | 3.38 | 0.68 |
| FlexiCodec @8.3Hz | 0.15 | 0.85 / 8q | 216M | 2.98 | **2.28** | 3.03 | 4.21 | 3.10 | 0.78 |
| ≤0.7kbps Acoustic Bitrate | | | | | | | | | |
| TAAE-25Hz-700bps | - | 0.70 / 2q | 953M | - | 4.19 | 2.84 | 4.33 | 4.98 | 0.58 |
| TAAE-25Hz-400bps | 0.40 | 0.40 / 1q | 953M | 5.21 | 5.21 | 2.61 | **4.34** | 4.10 | 0.53 |
| SemantiCodec-50Hz | 0.68 | 0.68 / 1q | 921M | 4.97 | 4.97 | 2.38 | 3.03 | 4.94 | 0.63 |
| SemantiCodec-25Hz | 0.34 | 0.34 / 1q | 1878M | 23.8 | 23.8 | 1.89 | 2.93 | 5.92 | 0.40 |
| TS3Codec(X4)-40Hz | 0.68 | 0.68 / 1q | 204M | 5.14 | 5.14 | 2.58 | 3.67 | 3.65 | 0.63 |
| TaDiCodec-6.25Hz | 0.15 | 0.15 / 1q | 751M | 4.32 | 4.32 | 1.73 | 4.05 | 9.75 | **0.83** |
| FlexiCodec @6.25Hz | 0.11 | 0.64 / 8q | 216M | **4.15** | **2.53** | 2.76 | 4.18 | **3.42** | 0.71 |

## 5 MORE EXPERIMENTS

• **Downstream TTS experiments.** In Appendix B, we detail our low-frame-rate TTS system with FlexiCodec (FlexiCodec-TTS). Our conclusions are (1) FlexiCodec-TTS achieves competitive performance with significant speedups over baselines, and (2) a higher NAR frame rate is important for high audio quality, but using lower frame rates for AR does not necessarily degrade performance.

• **Downstream audio understanding experiments.** In Appendix C, we show that FlexiCodec semantic token embeddings surpass other codecs in downstream audio understanding tasks, highlighting the potential to use FlexiCodec in unified multimodal understanding and generation frameworks.

• **Ablation study on other components of FlexiCodec.** In Appendix D, we confirm our other design choices like utilizing an ASR feature, transformer-based frame merging and unmerging modules, and FSQ quantization are beneficial to our codec's semantic and acoustic performance.

• **Codec efficiency analysis.** In Appendix E, we benchmark the encoding and decoding speed of FlexiCodec. The results show that FlexiCodec has a low Real-Time Factor (RTF) of 0.018 for encoding and 0.006 for decoding across all frame rates. This shows that our model is a practical and efficient solution for integration into larger pipelines like TTS systems.

• **Generalization to out-of-domain and multilingual data.** In Appendix F, we evaluate the robustness of our English-trained model on the diverse, out-of-domain Emilia (He et al., 2024) dataset. The results show that (1) FlexiCodec maintains strong semantic and acoustic performance on out-of-domain English speech, confirming it is not overfitted to audiobook data. (2) While the semantic tokens struggle on unseen languages in a zero-shot setting, the full acoustic reconstruction remains competitive with multilingual models. (3) A fine-tuning on an unseen language, Chinese, can adapt FlexiCodec's semantic tokens and the merging scheme to the new language.

## 6 CONCLUSION

In this work, we introduced FlexiCodec, a novel neural audio codec designed for very low frame rate operation. By incorporating a dynamic frame rate, an ASR-assisted dual-stream architecture, and Transformer-based frame merging/unmerging modules, FlexiCodec effectively preserves semantic information at low rates. Experiments confirm its strong performance in low frame rate and low bitrate speech coding, controllable frame rate, and effectiveness in a downstream TTS system. Limitaion and future work are discussed in Appendix J.

## 7 REPRODUCIBILITY STATEMENT

To ensure the reproducibility of our work, we provide comprehensive details on methodology, training configurations, and evaluation setup throughout the paper and appendices. All datasets used, including Librilight-Large, LibriSpeech, and TIMIT, are publicly available. We have also released code and model in the Github repository mentioned in the abstract.

## 8 ACKNOWLEDGEMENTS

This work is supported by the NSFC Grant 62376237; the Shenzhen Science and Technology Program ZDSYS20230626091302006; 2023 Shenzhen Stability Science Program; Program for Guangdong Introducing Innovative and Entrepreneurial Teams 2023ZT10X044.

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

## A    Visualization of Frame Merging Patterns

Figure 5 presents visualizations of FlexiCodec tokens aligned with the TIMIT (Garofolo et al., 1993) dataset phonemes. We show four randomly selected utterances from different speakers, each speaking the same content. For each utterance, the inferred content of merged tokens is labeled in green above the token sequence. Our discussions are as follows.

• **Typical merged frames correspond to phonemically coherent units.**    Frames forming syllables, short or common words (e.g., "su," "your," "all"), long vowels (e.g., "/aa/" in "dark", "water"), and silence intervals ([SIL]) are frequently merged to form single tokens spanning multiple original frame indices.

• **Merging patterns exhibit strong consistency across different speakers for the same utterance.** Although the four speakers differ in voice quality and prosody, the dynamic tokenization captures similar merging structures. For example, phonetic patterns of "she", "had", "su", "grea", "year" and the trailing silence are common to all four utterances.

• **The deterministic nature of merging allows interpretability and reproducibility.**    Since merging boundaries are derived from pretrained ASR features without additional trainable parameters, the token boundaries are replicable and interpretable. In this work, we reused the same ASR feature for both encoding semantic tokens and to derive the merging boundaries. We leave the exploration of more features as future works.

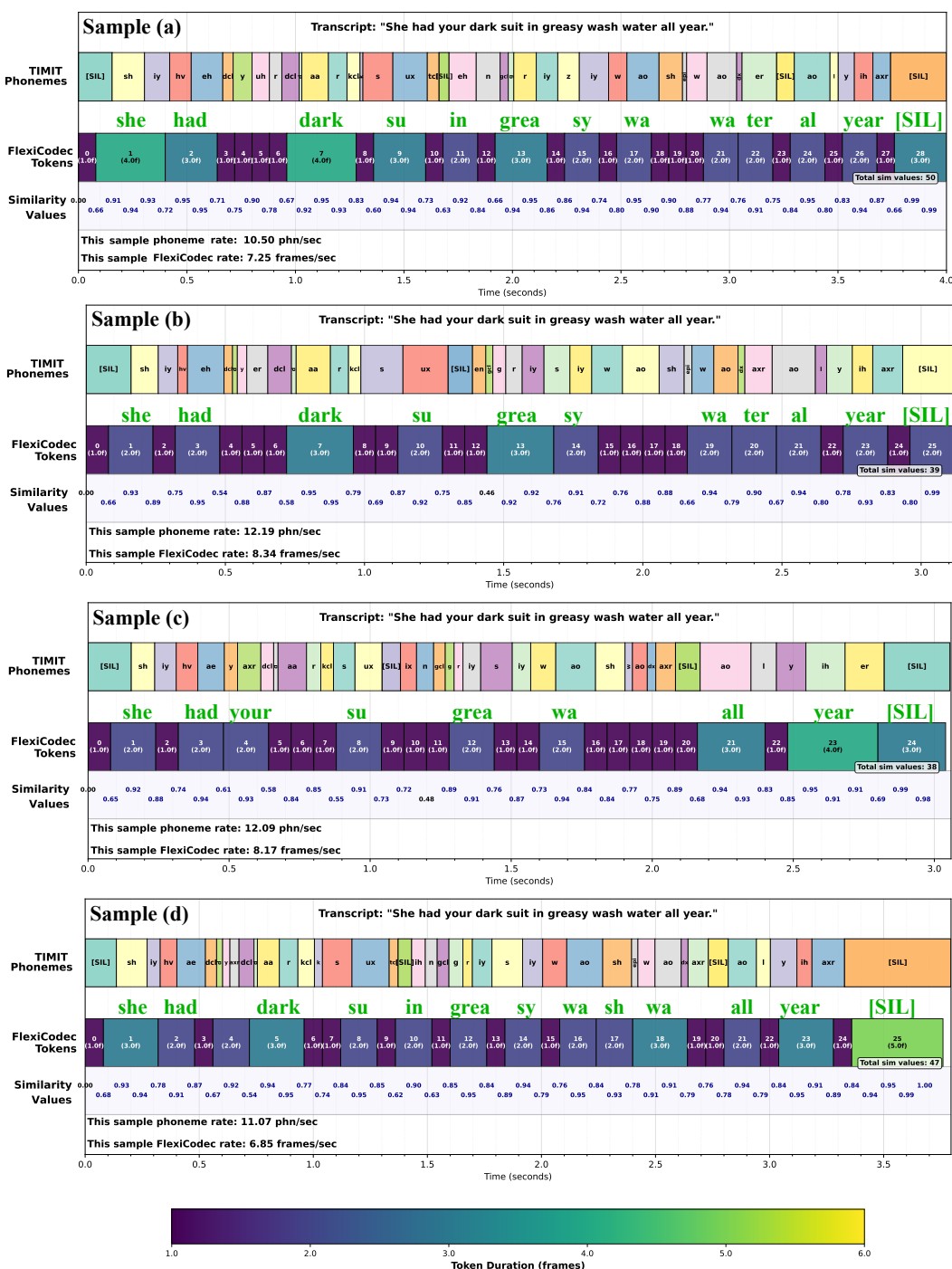

Figure 5: Visualizations of FlexiCodec dynamic-rate tokens aligned with TIMIT dataset phonemes. We visualize four random utterances of different speakers speaking the same transcript "She had your dark suit in greasy wash water all year." The inferred content of each merged token is labeled in green font.

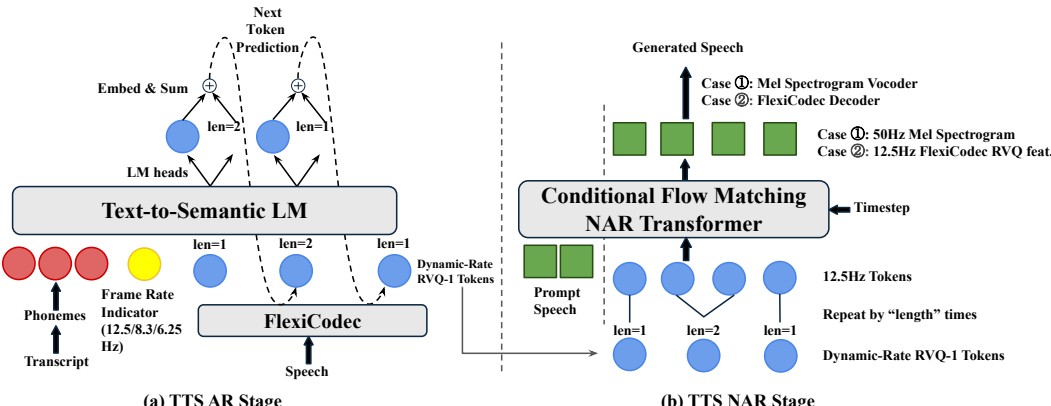

Figure 6: Overview of FlexiCodec-TTS. (a) The AR LM has two lightweight prediction heads to predict FlexiCodec RVQ-1 token and corresponding frame lengths. (b) The NAR Stage predicts continuous features conditioned on stage 1 outputs. We experiment with 2 types of continuous features, ① 50Hz Mel Spectrogram, and ② 12.5Hz FlexiCodec feature.

## B DOWNSTREAM EXPERIMENT: TTS

### B.1 TTS MODEL ARCHITECTURE

We integrate FlexiCodec within an AR+NAR(Flow Matching) TTS framework. Our model architecture is inspired by CosyVoice (Du et al., 2024a). Modifications have been made in the AR and NAR stages mainly because FlexiCodec produces dynamic frame rate tokens.

**AR Stage**  As shown in Figure 6(a), this is a Transformer AR model that predicts FlexiCodec RVQ1 tokens. Different from previous systems, our model has two token prediction heads instead of one. This is because each FlexiCodec token includes an explicit "frame length" attribute. Therefore, we treat **duration modeling as an auxiliary classification task** by adding a second prediction head after the last hidden layer. Each prediction head is a lightweight linear projector. The LM input sequence is:

$$\left[ \text{Ⓢ}, \{\mathbf{y}_u\}_{u \in [1:U]}, \text{Ⓜ}, \{\text{Embed}(q_t^{(s)}) + \text{Embed}(\ell_{t-1})\}_{t \in [1:\hat{T}]}, \text{Ⓔ} \right]$$

where Ⓢ, Ⓜ and Ⓔ denote the start, mid, and end separators. $\{\mathbf{y}_u\}_{u \in [1:U]}$ denotes the embedded phoneme sequence, $q_t^{(s)}$ is the $t$-th FlexiCodec RVQ1 token, and $\ell_{t-1}$ is the frame length of the $t-1$-th RVQ1 token. For training, we employ next-token prediction with two parallel classification heads attached to the LM's last hidden state, to predict both $q_t^{(s)}$ and $\ell_{t-1}$. The composite loss $\mathcal{L}$ is calculated as $\mathcal{L} = \mathcal{L}_{\text{sem}} + \lambda \mathcal{L}_{\text{dur}}$, where:

$$\mathcal{L}_{\text{sem}} = -\sum_{t=1}^{\hat{T}} \log P\big(q_t^{(s)} \mid p_{1:L}, q_{1:t-1}^{(s)}\big), \quad \mathcal{L}_{\text{dur}} = -\sum_{t=1}^{\hat{T}} \log P\big(\ell_t - 1 \mid p_{1:L}, q_{1:t-1}^{(s)}\big).$$

During inference, the model is prefixed with the phoneme sequence and a prompt speech's RVQ1 (tokens and frame lengths). The remaining tokens and frame lengths are sampled autoregressively.

**Frame Rate Flexibility in AR stage**  To enhance flexibility, the AR model is trained on sequences with either 12.5Hz, 8.3Hz, or 6.25Hz average frame rate, by setting various $\tau$ values when obtaining the RVQ1 tokens. To distinguish these sequences, a dedicated control token is prepended to the first audio token. During inference, the user can select from one of the three control tokens to specify an output frame rate.

**NAR Stage**  In this stage, we aim to transform the predicted RVQ-1 sequence into audios with acoustic details. While previous works like VALL-E (Wang et al., 2023) and SoundStorm (Borsos

et al., 2023b) have used NAR models to predict RVQ codec tokens, we have opted for predicting continuous features because low frame rate discrete tokens' acoustic quality upper bound is not as good as high frame rate (previously shown in Table 5). Following other works (Le et al., 2024; Du et al., 2024a; Zhang et al., 2025d), we employ a conditional flow matching (Lipman et al., 2022). This allows us to experiment with two acoustic features with different frame rates, (1) the 50Hz Mel spectrogram (100 Mel bands following (Zhang et al., 2025d)), and (2) the 12.5Hz FlexiCodec feature (the feature labeled in dark green in Figure 1). For (2), it is equivalent to predicting $n - 1 = 23$ layers of RVQ-rest tokens simultaneously.

As shown in Figure 6, the model is a flow matching transformer. During training, given a speech waveform $\mathbf{u}$ and its acoustic feature $\mathbf{y}_1$, we randomly select a prefix of $\mathbf{y}_1$, denoted as $\mathbf{y}_1^{ctx}$, and aim to reconstruct the other part (denoted as $\mathbf{y}_1^{mis}$) conditioned on $\mathbf{y}_1^{ctx}$ and the FlexiCodec RVQ-1 tokens. To obtain the conditioning FlexiCodec RVQ-1 tokens, we first expand the dynamic-frame-rate RVQ-1 sequence $q_{1:\hat{T}}^{(s)}$ into 12.5Hz fixed frame rate sequence $q_{1:T}^{(s)}$ using frame lengths $\ell_{1:\hat{T}}$, and then linearly interpolat it to match the frame rate of $\mathbf{y}_1$. The rest of the formulation follows Zhang et al. (2025d); Du et al. (2024a), using a 300M Diffusion Transformer to approximate an optimal transport path between gaussian noise $\mathbf{y}_0$ and spectrogram $\mathbf{y}_1$. Classifier-free guidance (CFG) is applied by randomly dropping conditions during training and combining unconditional and conditional predictions during inference. During inference, we use 15 timesteps, and a CFG guidance strength of 1.5. To decode mel spectrogram, we use a pretrained Vocos (Siuzdak, 2023) vocoder from the Amphion (Li et al., 2025b; Zhang et al., 2023b) toolkit.

**Frame Rate Flexibility in NAR stage**   The NAR model supports decoding from either 12.5Hz, 8.3Hz, or 6.25Hz FlexiCodec RVQ-1 sequence. Its conditioning FlexiCodec RVQ-1 sequences are expanded from either 12.5Hz, 8.3Hz, or 6.25Hz average frame rate, by randomly choosing from corresponding $\tau$ values during training. We do not add additional tokens to distinguish these sequences because we find it yields the same results.

## B.2   TTS EXPERIMENTS

**TTS Training Configuration**   We use the 16kHz, 50k-hour Libriheavy (Kang et al., 2024) dataset for training. Each model is trained for 600k steps on 8 Nvidia V100 32GB GPUs. The AR model uses AdamW (Loshchilov & Hutter, 2017) optimizer with lr=$1 \times 10^{-4}$, betas=$(0.5, 0.999)$; batch size is 96 seconds per GPU; gradient accumulation step is 3. The NAR models uses lr=$7.5 \times 10^{-5}$, betas=$(0.5, 0.999)$; batch size is 24 seconds per GPU; gradient accumulation step is 3. The AR model has 250M parameters; the NAR model has 300M parameters.

**TTS Evaluation Metrics**   We use the E2TTS (Eskimez et al., 2024) test suite[4] to test our models. Its test data consists of 1,132 (reference, test) pairs from Librispeech-PC-test-clean (Meister et al., 2023). We evaluate both objective and subjective metrics. The objective metrics include word error rate (WER) and speaker similarity (SIM-o). We report the real-time factor (RTF) of each system tested on an RTX5000 GPU. RTF is calculated as the ratio of the total processing time to the total duration of generated audios. The WER metric uses a Hubert-Large-LS960-ft ASR system (Hsu et al., 2021), and the SIM-o calculates the cosine similarity of speech samples of each test utterance against reference clips using a WavLM-large-based speaker verification model. For subjective evaluations, we evaluate Naturalness Mean Opinion Score (NMOS) and Quality Mean Opinion Score (QMOS). For each MOS, six professional linguistic experts evaluate 10 audios per system. The human evaluation instructions are attached in Appendix L. Baseline systems include E2TTS (Eskimez et al., 2024), VALL-E (Wang et al., 2023), CosyVoice (Du et al., 2024a), and SparkTTS (Wang et al., 2025b), FireRedTTS (Guo et al., 2024).

**TTS Evaluation Results**   We present the evaluation results in Table 6. Our analysis is as follows:

• **FlexiCodec-TTS achieves competitive performance with significant speedups.** The FlexiCodec-TTS model with a 50Hz NAR decoder demonstrates highly competitive performance across both objective and subjective metrics. At its 6.25Hz AR setting, it achieves a WER of 3.2%, Naturalness MOS (NMOS) of 3.32 and a Quality MOS (QMOS) of 3.40, rivaling or exceeding

---

[4]https://github.com/microsoft/e2tts-test-suite

Table 6: TTS evaluation results. Systems with * are borrowed from official results. Each FlexiCodec-TTS is configured to work on each frame rate option without retraining. The actual average frame rate of FlexiCodec-TTS is calculated on all the generated audios during testing.

| Model | AR Frame rate | WER↓ | SIM-o↑ | RTF(AR)↓ | RTF(AR+NAR)↓ | NMOS↑ | QMOS↑ |
|---|---|---|---|---|---|---|---|
| E2TTS* | 94 (NAR) | **2.0** | **0.68** | - | - | - | - |
| VALL-E* | 75 | 4.9 | 0.50 | | - | - | - |
| CosyVoice | 50 | 3.2 | 0.63 | 0.51 (7.3X) | 0.62 (3.4X) | **3.17±0.95** | 3.32±0.85 |
| SparkTTS | 50 | 6.0 | 0.55 | 0.83 (11X) | 0.84 (4.7X) | 3.15±0.87 | **3.47±0.74** |
| FireRedTTS | 25 | 6.8 | 0.43 | **0.35 (5X)** | **0.47 (2.6X)** | 2.92±0.95 | 3.10±0.81 |
| FlexiCodec-TTS (w/ 50Hz NAR) | 12.5 (12.5 actual) | **2.5** | 0.64 | 0.15 (2.1X) | 0.26 (1.4X) | 3.27±0.95 | 3.30±0.84 |
| | 8.3 (8.0 actual) | **2.5** | **0.65** | 0.10 (1.4X) | 0.22 (1.2X) | 3.22±0.91 | 3.28±0.84 |
| | 6.25 (6.3 actual) | 3.2 | **0.65** | **0.07 (1X)** | **0.18 (1X)** | **3.32±0.87** | **3.40±0.78** |
| FlexiCodec-TTS (w/ 12.5Hz NAR) | 12.5 (12.5 actual) | **3.1** | **0.51** | 0.15 (2.1X) | 0.19 (1.1X) | 2.87±0.90 | 2.92±0.76 |
| | 8.3 (8.0 actual) | 3.6 | 0.50 | 0.10 (1.4X) | 0.13 (0.7X) | 2.92±0.99 | **2.95±0.88** |
| | 6.25 (6.3 actual) | 4.6 | 0.48 | **0.07 (1X)** | **0.10 (0.6X)** | **2.93±1.06** | 2.87±0.74 |

strong baselines like CosyVoice (WER 3.2%, NMOS 3.17, QMOS 3.32) and SparkTTS (WER 6.0%, NMOS 3.15, QMOS 3.47). Crucially, this high perceptual quality is achieved with a significant reduction in computational cost. The 6.25Hz AR configuration has an RTF of just 0.07 for the autoregressive stage, representing a 7.3x speedup over CosyVoice's AR model. Adding the inference time of 50Hz NAR, the total speedup is also substantial, at 3.4x.

FlexiCodec-TTS using 12.5Hz NAR shows degraded performance, but still outperforms baselines like VALL-E and FireRedTTS, while achieving higher speedups. This balance of speed and quality makes FlexiCodec-TTS highly suitable for low-latency, resource-constrained applications.

• **A higher NAR frame rate is important for high audio quality, but using lower frame rates for AR does not necessarily degrade performance.** A comparison between our two FlexiCodec-TTS configurations reveals the distinct roles of the AR and NAR stages. Using a 50Hz NAR decoder consistently yields superior results over a 12.5Hz NAR across all metrics, including WER, SIM-o, NMOS, and QMOS. This confirms that a higher temporal resolution in the acoustic decoding stage is vital for reconstructing fine-grained details and preserving naturalness.

Interestingly, for the AR stage, lower frame rates do not always degrade performance. With 50Hz NAR, the 8.3Hz and 6.25Hz AR configurations achieve comparable or even slightly better WER (2.5 vs. 2.5 vs. 3.2), SIM-o (0.64 vs. 0.65 vs. 0.65) compared to 12.5Hz AR. To explain this phenomenon, we suggest that (1) lower frame rate tokens might have better semantic and acoustic information disentanglement, and (2) a shorter sequence may alleviate the challenge of modeling long-range dependencies, allowing the Transformer to learn more effective attention patterns.

## C  DOWNSTREAM EXPERIMENT: AUDIO UNDERSTANDING

Table 7: Evaluation on various audio understanding tasks.

| Model \ Dataset | Spoofing detection | Emotion recog. | Sound event detection | Environment classification | Keyword spotting | Speaker counting | Emotion recog. | Non-speech sounds | Language identif. |
|---|---|---|---|---|---|---|---|---|---|
| | ASVSpoof2015 | CREMA-D | DESED | ESC50 | Fluent speech commands | Libricount | RAVDESS | Vocalsound | Voxlingua33 |
| *Continuous feature* | | | | | | | | | |
| Whisper | **0.97** | 0.57 | 0.13 | 0.53 | 0.78 | 0.55 | 0.46 | 0.86 | **0.87** |
| Dasheng | **0.97** | **0.76** | **0.53** | **0.86** | 0.95 | **0.68** | **0.75** | **0.91** | 0.81 |
| SenseVoice-Small | 0.95 | 0.71 | 0.29 | 0.59 | **0.99** | 0.49 | 0.68 | 0.89 | 0.81 |
| *Codec models w/ RVQ-1 token features* | | | | | | | | | |
| WavTokenizer-75Hz | **0.95** | 0.33 | 0.16 | 0.19 | 0.02 | 0.38 | 0.27 | 0.27 | 0.09 |
| Mimi-12.5Hz | 0.93 | 0.46 | 0.09 | 0.33 | 0.59 | 0.49 | 0.35 | 0.73 | 0.30 |
| DualCodec-12.5Hz | 0.93 | 0.35 | 0.11 | **0.35** | 0.82 | **0.52** | 0.33 | 0.68 | 0.64 |
| FlexiCodec @12.5Hz | 0.93 | **0.58** | 0.17 | 0.30 | **0.99** | 0.43 | **0.50** | **0.79** | **0.72** |
| FlexiCodec @8.3Hz | 0.92 | **0.58** | **0.18** | 0.28 | **0.99** | 0.43 | 0.50 | 0.73 | 0.69 |
| FlexiCodec @6.25Hz | 0.92 | 0.54 | 0.17 | 0.27 | 0.98 | 0.42 | 0.49 | 0.74 | 0.67 |

To evaluate the quality of the semantic tokens produced by FlexiCodec, and to validate whether FlexiCodec is suitable for building an audio language model, we test its RVQ-1 token embeddings on a diverse suite of 9 audio understanding tasks from the XARES benchmark (Zhang et al., 2025c).

We compare against two types of representations: (1) continuous features from strong speech SSL or ASR models (Whisper (Radford et al., 2023), Dasheng (Dinkel et al., 2024), and SenseVoice-Small (An et al., 2024)), and (2) RVQ-1 tokens from other audio codecs. For each task, we train a simple downstream model consisting of a linear classifier on top of the frozen embeddings. The results are presented in Table 7.

The results show that continuous features from large speech models generally outperform codec tokens, which is expected given their continuous nature. Among the codec models, FlexiCodec demonstrates superior performance across most tasks. Notably, on keyword spotting, FlexiCodec at 12.5Hz and 8.3Hz achieves a score of 0.99, matching the performance of the best continuous feature model, SenseVoice-Small, and significantly outperforming other codecs. This highlights the rich semantic information captured in FlexiCodec's tokens. Furthermore, FlexiCodec excels in emotion recognition, non-speech sound detection, and language identification, consistently surpassing other codec models. Even at a very low average frame rate of 6.25Hz, FlexiCodec maintains strong performance, demonstrating its efficiency and effectiveness in preserving crucial semantic content. These results validate that FlexiCodec's tokens are not only suitable for high-quality audio reconstruction but also serve as a powerful and compact representation for general audio understanding.

# D    ABLATION STUDY AND ANALYSIS

## D.1    ABLATION STUDY

Table 8: Extended ablation study of FlexiCodec. We underline the results that are significantly degraded from the baseline.

| Label | Ablated Item | Codec Param | Semantic Test | | Acoustic Test (RVQ1:8) | | | |
|---|---|---|---|---|---|---|---|---|
| | | | WER(RVQ1)↓ | WER(RVQ1:8)↓ | PESQ↑ | UTMOS↑ | MCD↓ | SIM↑ |
| *Baseline* | | | | | | | | |
| A1 | FlexiCodec @6.25Hz | 216M | 4.15 | 2.53 | 2.76 | 4.18 | 3.42 | 0.71 |
| *Frame merging module* | | | | | | | | |
| B1 | w/o Transformer in this module | 176M | 4.19 | 2.72 | 2.46 | 4.08 | 3.78 | 0.67 |
| B2 | Use shared params for the two streams | 216M | 8.24 | 2.37 | 2.78 | 4.20 | 3.43 | 0.70 |
| *Frame Unmerging module* | | | | | | | | |
| C1 | w/o Transformer in this module | 116M | 4.22 | 3.15 | 2.56 | 4.05 | 3.65 | 0.70 |
| *Quantization* | | | | | | | | |
| D1 | Use VQ instead of FSQ | 216M | 4.43 | 2.66 | 2.74 | 4.18 | 3.48 | 0.68 |
| *Semantic feature* | | | | | | | | |
| E1 | Use w2v-bert-2 SSL feat. | 216M | 27.3 | 5.88 | 2.47 | 3.90 | 3.62 | 0.74 |
| E2 | Use Whisper-medium feat. | 216M | 9.83 | 3.27 | 2.69 | 4.11 | 3.50 | 0.69 |
| E3 | Use avg SenseVoice-Small feat. | 216M | 4.66 | 2.55 | 2.74 | 4.18 | 3.44 | 0.73 |

Table 8 summarizes our further ablation study of FlexiCodec, examining the impact of removing or modifying key components including the Frame Merging and Unmerging modules, quantization schemes, and semantic feature inputs. Each ablated model is retrained following the recipe of FlexiCodec and are tested on 6.25Hz average frame rates. Key observations are summarized below.

• **Transformers in Frame Merging and Unmerging Modules are critical for maintaining high acoustic quality.**    Removing the Transformer refinement from either the frame merging or unmerging module leads to noticeable decreases in acoustic quality metrics (PESQ, UTMOS) and speaker similarity (SIM), as evidenced by the underlined values in rows B1 and C1. However, the impact on the semantic test scores, especially RVQ-1 WER, is very small. These results highlight that the Transformer's local attention especially refines acoustic features after compression and expansion, **reducing artifacts and unnatural transitions caused by naive averaging and frame repetition.**

• **Using an SSL feature does not yield a good performance in FlexiCodec, compared to ASR features.** As shown in row E1, adopting the w2v-bert-2 (Barrault et al., 2023) SSL feature (which worked in DualCodec) in our system shows degraded semantic and acoustic results, except that its SIM score improves. We have examined the frame similarity pattern of this feature, and found that its adjacent frames tend to have low similarities, as it required to set a very low threshold

$\tau \approx 0.5$ to reduce to a 6.25Hz average frame rate. We hypothesize that the SSL feature contains leaked acoustic information like timbre, so the frame merging module may not effectively merge semantically consistent tokens together. As a comparison, we also tried using Whisper-medium ASR (row E2) and layer-averaged Sensevoice-Small ASR (row E3). Both perform better than SSL feature, suggesting that the semantic-rich ASR features are more suitable for dynamic-frame-rate codecs like FlexiCodec.

## D.2 FACTORIAL ANALYSIS OF FLEXICODEC DESIGN CHOICES

Table 9: Factorial analysis of FlexiCodec design choices in relation to RVQ1 WER at 6.25Hz average frame rate.

| Label | FlexiCodec Design choice | | | | WER(RVQ1) ↓ | Source of data |
| | Semantic feat | Dynamic rate? | Merging&Unmerging transformer? | FSQ? | | in this manuscript |
|---|---|---|---|---|---|---|
| F1 | w2v-bert-2 SSL | ✗ | ✗ | ✗ | 31.5 | Figure 3 (DualCodec) |
| F2 | SenseVoice ASR | ✗ | ✗ | ✗ | 5.99 | Section 4.2 (in text) |
| F3 | SenseVoice ASR | ✗ | ✓ | ✗ | 5.40 | - |
| F4 | SenseVoice ASR | ✓ | ✓ | ✗ | 4.43 | Table 8 (Ablation D1) |
| F5 | SenseVoice ASR | ✗ | ✓ | ✓ | 5.22 | Table3 |
| F6 | SenseVoice ASR | ✓ | ✓ | ✓ | **4.15** | Table 3&5 (FlexiCodec) |

Table 9 presents a factorial analysis to examine the impact of four central components: the semantic feature source, the Merging&Unmerging transformers, the dynamic rate mechanism, and Finite State Quantization (FSQ). We note that 5 of the 6 data points in the table have occured previously in the manuscript, so the table mainly provides a factorial view of the data. We compare at an average frame rate of 6.25Hz to isolate the incremental lift of each design choice on semantic preservation, as measured by RVQ1 reconstruction WER. Key observations are summarized below:

• **Changing from SSL to ASR feature is the primary driver for semantic preservation at a fixed low frame rate.** As shown by comparing F1 and F2 in Table 9, the most significant improvement comes from replacing the DualCodec-style w2v-bert-2 SSL features with SenseVoice ASR features. This single change causes the RVQ1 WER to decrease from 31.5 to 5.99. This confirms that using a more semantic-rich feature derived from ASR is the foundational step for achieving good semantic representation at a fixed low rate.

• **The dynamic rate mechanism is key to further improving semantic preservation.** As discussed in Section 4.3, the dynamic frame rate mechanism is key to further improving semantic information while allowing controllable frame rates. This can be confirmed by examining F3 vs. F4, and F5 vs. F6. We note that the dynamic frame merging reuses the computed ASR features, thus assumes the benefits of the semantic-rich ASR features.

• **The Merging Unmerging transformers and FSQ provide incremental benefits.** The transformers and FSQ also contribute to the final performance. Adding the 140M-parameter of the transformers (F2 vs. F3) reduces WER from 5.99 to 5.40, showing that the transformers can also benefit fixed-frame-rate codecs. Similarly, integrating FSQ into the full model (F4 vs. F6) provides the final refinement, lowering the WER from 4.43 to our best result of 4.15. Overall, our analysis validates that all components work in concert, but the combination of ASR features and our dynamic rate mechanism delivers the core innovation and performance gains.

## E THE SPEED OF FLEXICODEC

Table 10 presents a comparison of the encoding and decoding speeds, measured by Real-Time Factor (RTF), for FlexiCodec and other state-of-the-art neural codecs. RTF is measured by the ratio of processing time to audio duration. All models were benchmarked on one Nvidia V100 GPU using fp32 precision with a batch size of 1. The average RTF was calculated over our cropped LibriSpeech-test-clean test set.

The results show that, while FlexiCodec is not the most efficient among the evaluated systems, it still demonstrates acceptable performance. Its encoding and decoding RTF remains constant at 0.018

Table 10: Encoding and decoding speed comparison. All models are run on a V100 GPU with fp32 precision and a batch size of 1. We report the average Real-Time Factor (RTF) on our test set.

| Model | Encode Params | Encode RTF↓ | Decode Params | Decode RTF↓ |
|---|---|---|---|---|
| DAC-official-75Hz | 22M | 0.004 | 52M | **0.002** |
| Encodec-75Hz | 7.4M | 0.004 | 7.4M | 0.004 |
| SpeechTokenizer-50Hz | 68M | 0.005 | 35M | 0.007 |
| WavTokenizer-75Hz | 8.8M | 0.005 | 62M | **0.002** |
| Mimi-12.5Hz | 38M | **0.003** | 40M | **0.002** |
| SemantiCodec-50Hz | 725M | 0.100 | 196M | 0.650 |
| SemantiCodec-25Hz | 1660M | 0.120 | 218M | 0.643 |
| XYTokenizer-12.5Hz | 306M | 0.022 | 214M | 0.005 |
| DualCodec-12.5Hz | 622M | 0.019 | 53M | **0.002** |
| FlexiCodec@12.5Hz | 289M | 0.018 | 161M | 0.006 |
| FlexiCodec@8.3Hz | 289M | 0.018 | 161M | 0.006 |
| FlexiCodec@6.25Hz | 289M | 0.018 | 161M | 0.006 |

(encode) and 0.006 (decode) across different frame rate options (12.5Hz, 8.3Hz, and 6.25Hz). The low RTF values indicate that FlexiCodec only adds a very small overhead to downstream tasks like TTS, whose RTFs are usually more than 0.1. This positions FlexiCodec as a practical and scalable solution for real-world applications.

# F   TESTING FLEXICODEC ON OUT-OF-DOMAIN AND MULTILINGUAL DATA

As FlexiCodec is trained on an audiobook dataset (LibriLight-Large), we wish to know its performance on out-of-domain (OOD) and multilingual speech data to assess its generalizability. For this evaluation, we use Emilia (He et al., 2024), an in-the-wild multilingual dataset featuring spontaneous and conversational speech. As Emilia does not provide an official test set, we craft our test cases by randomly choosing audios from its training set. For English, we choose one audio from each shard, forming 1,140 speech samples. For other languages, we randomly select 200 samples per language. For each language, we use whisper-large-v3 (Radford et al., 2023) and evaluate against the transcripts in Emilia dataset's metadata. For FlexiCodec evaluations, we determine $\tau$ values from trial runs on each tested language.

Table 11: Comparison between neural audio codecs on English Emilia dataset. We use 1,140 randomly sampled speech from Emilia-EN subset. Emilia contains diverse spontaneous speech which is out of domain (OOD) for FlexiCodec.

| System | RVQ1 BR(kbps) | RVQ1:8 BR(kbps)/n_q | Is OOD? | Semantic Test | | Acoustic Test (RVQ1:8) | | | |
|---|---|---|---|---|---|---|---|---|---|
| | | | | WER(RVQ1)↓ | WER(RVQ1:8)↓ | PESQ↑ | UTMOS↑ | MCD↓ | SIM↑ |
| *>1kbps Acoustic Bitrate* | | | | | | | | | |
| DAC-75Hz | 0.75 | 6.0 / 8q | ✓ | 34.7 | 6.26 | 3.67 | 2.95 | 3.26 | 0.91 |
| Encodec-75Hz | 1.50 | 6.0 / 8q | ✓ | 10.3 | 6.30 | 3.10 | 2.50 | 3.59 | **0.92** |
| SpeechTokenizer-50Hz | 0.50 | 4.0 / 8q | ✓ | 26.8 | 7.33 | 2.70 | 3.06 | 4.77 | 0.84 |
| Mimi-12.5Hz | - | 1.1 / 8q | ✓ | - | 9.37 | 2.69 | 3.11 | 5.14 | 0.78 |
| DualCodec-12.5Hz | 0.19 | 1.2 / 8q | ✗ | 14.2 | 6.41 | **3.17** | 3.47 | **4.09** | 0.89 |
| XYTokenizer | - | 1.0 / 8q | ✗ | - | **6.20** | 2.92 | 3.35 | 4.46 | 0.89 |
| FlexiCodec @12.5Hz | 0.23 | 1.3 / 8q | ✓ | **8.25** | 6.47 | 3.01 | **3.61** | 4.18 | 0.84 |
| *~0.8kbps Acoustic Bitrate* | | | | | | | | | |
| WavTokenizer-75Hz | 0.90 | 0.90 / 1q | ✓ | 10.7 | 10.7 | **2.73** | 3.40 | 4.80 | 0.75 |
| XCodec2-50Hz | 0.80 | 0.80 / 1q | ✗ | **8.00** | 8.00 | 2.23 | 2.70 | 6.88 | 0.76 |
| FlexiCodec @8.3Hz | 0.15 | 0.85 / 8q | ✓ | 10.8 | **6.88** | 2.64 | **3.59** | **4.77** | **0.77** |
| *≤0.7kbps Acoustic Bitrate* | | | | | | | | | |
| TAAE-25Hz-700bps | - | 0.70 / 2q | ✓ | 17.3 | 17.3 | **2.47** | **3.90** | 6.57 | 0.56 |
| SemantiCodec-50Hz | 0.68 | 0.68 / 1q | ✓ | 13.6 | 13.6 | 2.13 | 2.35 | 6.60 | 0.61 |
| FlexiCodec @6.25Hz | 0.11 | 0.64 / 8q | ✓ | **12.9** | **7.76** | 2.36 | 3.51 | **5.30** | **0.70** |

**Testing on English Out-of-Domain Data**   As shown in Table 11, FlexiCodec obtains competitive performance in out-of-domain English speech. In each bitrate category, FlexiCodec achieves competitive results especially in semantic preservation. On the English Emilia subset, the WER (RVQ1)

Table 12: Multilingual Evaluation of codec systems. Among the compared systems. FlexiCodec and SpeechTokenizer are trained only on English speech.

| System | WER(RVQ1) ↓ | | | | | WER(RVQ1:8) ↓ | | | | | SIM(RVQ1:8) ↑ | | | | |
|---|---|---|---|---|---|---|---|---|---|---|---|---|---|---|---|
| | ZH | KO | JA | FR | DE | ZH | KO | JA | FR | DE | ZH | KO | JA | FR | DE |
| DAC-official-75Hz | 35.7 | 77.5 | 74.3 | 60.1 | 49.6 | **4.87** | 25.8 | 16.9 | 13.6 | 11.7 | **0.92** | **0.92** | **0.91** | **0.92** | **0.92** |
| SpeechTokenizer-50Hz | 100 | 100 | 100 | 100 | 72.7 | 6.26 | 29.6 | 18.2 | 15.8 | 13.4 | 0.84 | 0.83 | 0.81 | 0.82 | 0.86 |
| WavTokenizer-75Hz | **9.19** | **30.0** | **23.8** | **16.8** | **18.5** | 9.12 | 30.0 | 23.8 | 16.8 | 18.5 | 0.81 | 0.85 | 0.72 | 0.81 | 0.85 |
| Mimi-12.5Hz | - | - | - | - | - | 7.32 | 32.5 | 26.1 | 16.9 | 14.4 | 0.77 | 0.77 | 0.74 | 0.77 | 0.78 |
| XYTokenizer-12.5Hz | - | - | - | - | - | 7.28 | 30.0 | 20.2 | 16.8 | 18.5 | 0.91 | 0.89 | 0.83 | 0.81 | 0.85 |
| DualCodec-12.5Hz | 12.9 | 47.9 | 47.7 | 36.8 | 28.6 | 5.43 | 26.1 | 18.2 | 13.9 | 13.0 | 0.91 | 0.89 | 0.88 | 0.87 | 0.89 |
| FlexiCodec@12.5Hz | 100 | 100 | 58.5 | 97.5 | 52.0 | 6.29 | 29.6 | 18.0 | 14.5 | 13.4 | 0.83 | 0.83 | 0.82 | 0.82 | 0.85 |
| FlexiCodec@8.3Hz | 100 | 100 | 72.7 | 88.2 | 71.0 | 8.32 | 27.3 | 31.8 | 16.2 | 14.9 | 0.74 | 0.75 | 0.74 | 0.73 | 0.78 |
| FlexiCodec-ZH_tune@12.5Hz | 8.51 | 48.5 | 28.0 | 72.6 | 50.0 | 5.91 | 27.8 | 18.8 | 14.8 | 11.9 | 0.88 | 0.84 | 0.83 | 0.82 | 0.84 |
| FlexiCodec-ZH_tune@8.3Hz | 10.7 | 59.7 | 34.4 | 100 | 65.9 | 7.10 | 31.9 | 31.1 | 17.9 | 15.3 | 0.78 | 0.78 | 0.77 | 0.72 | 0.76 |

scores for FlexiCodec at 12.5Hz, 8.3Hz and 6.25Hz are 8.25%, 10.8%, and 12.9%. This performance is highly competitive with other state-of-the-art codecs (e.g., DualCodec-12.5Hz has 14.2%, WavTokenizer-75Hz has 10.7%), confirming that FlexiCodec's superior semantic preservation is not overfitted to clean, audiobook speech, but handles spontaneous, out-of-domain speech robustly. The acoustic results are robust to OOD data as well, showing competitive results to in-domain models like DualCodec and XYTokenizer.

**Testing on Multilingual Data**   To further probe the generalizability of FlexiCodec, we evaluate its performance on non-English languages from the Emilia dataset. The results are shown in Table 12. Since FlexiCodec is trained exclusively on English speech, this presents a significant OOD challenge. When examining the semantic preservation via WER(RVQ1), our model, along with other English-only models like SpeechTokenizer, struggles significantly, often resulting in high WERs. This is expected, as the semantic tokens are optimized for English phonetics and are not equipped to represent the nuances of other languages. However, when evaluating the more fine-grained WER(RVQ1:8) and SIM, which leverages the full acoustic information, FlexiCodec demonstrates a good ability to generalize. For instance, FlexiCodec @12.5Hz achieves WERs (e.g., 14.5 in French, 13.4 in German) that are competitive with multilingual models like XYTokenizer and Mimi. The acoustic similarity scores (SIM) also highlight FlexiCodec's cross-lingual generalization.

**Adapting FlexiCodec to Out-of-Domain Data**   We demonstrate that FlexiCodec can be adapted to OOD data by finetuning it with Chinese (ZH) data. To perform this adaptation, we continue training FlexiCodec with Chinese data from Emilia dataset for 250K steps. The training data does not overlap with the test set. Results are shown as "FlexiCodec-ZH_tune" in Table 12. After finetuning on Chinese data, it sees a dramatic improvement in semantic preservation for Chinese (ZH) at both 12.5Hz and 8.3Hz. This demonstrates that the semantic tokenizer of FlexiCodec can learn the phonetic characteristics of a new language with targeted training; the frame merging scheme applies across languages. Furthermore, it is seen that this adaptation also improves FlexiCodec's generalization ability to other languages, as the fine-tuned model shows notable performance gains in languages it was not explicitly trained on. Notably, "FlexiCodec-ZH_tune@12.5Hz" improves its WER(RVQ1) score on Japanese (JA) from 58.5% to 28.0% and on Korean (KO) from 100% to 48.5%.

# G   MORE INFORMATION OF BASELINE CODECS

## G.1   RETRAINED BASELINES

• **DAC:** It is a high-fidelity neural audio codec with open-source recipes. It proposed two key improvements over Encodec. First, it addressed the codebook collapse problem by reducing the dimension of the latent vector to a small value for quantization. Second, It replaced the ReLU activation function with a periodic activation function, offering benefits for reconstructing periodic signals such as speech and music. It is trained on more than 10k hours of data including speech, audio and music.

- **DualCodec:** It is a low-frame-rate neural audio codec specifically designed to enhance speech generation tasks like TTS. It introduced a dual-stream encoding architecture that processes semantic and acoustic information in parallel. One stream uses a pre-trained SSL model w2v-bert-2 (Barrault et al., 2023) to extract rich semantic features. The second stream encodes acoustic information directly from the waveform. This dual approach ensures that the primary codec tokens are semantically meaningful and allows low frame rates (12.5Hz or 25Hz). Its waveform encoding stream utilizes the architecture of DAC codec.

To retrain these two systems, we use their public repositories, and modify their codec encoder strides to achieve lower frame rates compared to their original versions. The configurations are shown in Table 13. Their decoders mirror their encoders. We have also modified their codebook sizes to align with FlexiCodec. Specifically, we modify to 32768 RVQ-1 codebook size for DualCodec instead of 16384, and use 4096 codebook size for DAC and DualCodec remaining RVQ layers.

To retrain fixed-frame-rate FlexiCodec baselines, we also follow Table 13 to modify their strides to achieve 8.3Hz and 6.25Hz frame rate, respectively.

Table 13: The CNN encoder strides settings corresponding to each frame rate configuration on retrained codec systems.

| Codec frame rate | Encoder stride settings |
|---|---|
| 12.5Hz | [4,4,5,8,2] |
| 8.3Hz | [4,5,6,8,2] |
| 6.25Hz | [4,5,8,8,2] |

## G.2 OTHER BASELINES

- **Encodec** (Défossez et al., 2022): A high-fidelity neural audio codec. It introduced LSTM bottleneck layers and multi-scale STFT discriminator on top of SoundStream (Zeghidour et al., 2021).

- **SpeechTokenizer** (Zhang et al., 2023a): A neural audio codec designed for speech language models. It introduced semantic distillation, where the first layer of its RVQ is trained to match semantic-rich representations from a pre-trained SSL model HuBERT.

- **Mimi** (Défossez et al., 2024): A neural audio codec designed for real-time applications. It compresses 24kHz speech into a 12.5 Hz token representation with a bitrate of 1.1 kbps. It features a split RVQ system, where one quantizer captures semantic information and the remaining layers encode acoustic details. It integrates Transformer layers in its bottleneck and distilling semantic knowledge from an SSL model.

- **WavTokenizer** (Ji et al., 2024): A codec with single-layer neural quantizer operating with a large FSQ codebook, optimized for universal speech processing and compression. It is noted for its efficiency and ease of integration into generative frameworks like GPT-4o.

- **TS3Codec** (Wu et al., 2024): A transformer-based neural audio codec with FSQ bottleneck, designed for minimal bitrate speech encoding. We acquire its inference audios from its audios. Its two model offerings, X2 and X4, have been described in its paper.

- **SNAC** (Siuzdak et al., 2024): A codec that extends the standard Residual Vector Quantization (RVQ) framework by implementing quantizers operating at different temporal resolutions (Siuzdak et al., 2024). Its coarse tokens are sampled at lower frequencies. Each RVQ layer operates at 12Hz, 23Hz, 47Hz, respetively.

- **TaDiCodec** (Wang et al., 2025c): A speech tokenizer with 6.25Hz tokens and text-assisted decoding, designed for text-to-speech synthesis. TaDiCodec has a transformer-based encoder and a diffusion-based decoder. Its diffusion decoder utilizes text transcriptions to enhance semantic; reference speech to enhance timbre. During inference, we use the ground truth audio as reference speech, and use HuBERT-Large-LS960-ft (Hsu et al., 2021) to obtain the text. We calculate its bitrate as speech token bitrate + text token theoretical bitrate.

• **TAAE (Stable-Codec)** (Parker et al., 2024): a transformer-based neural audio codec with FSQ or rFSQ bottlenecks. It aims to achieve high-quality speech at very low bitrates (0.4kbps and 0.7kbps). The model is effectively scaled up to 1B parameters.

• **SemantiCodec** (Liu et al., 2024): a neural audio codec with a diffusion decoder, operating at very low bitrates between 0.31 kbps and 1.43 kbps. Its training data incorporates speech, sound and music.

• **XYTokenizer** (Gong et al., 2025): A dual-channel 12.5Hz neural speech codec with parallel semantic and acoustic processing streams, operating at low bitrates. The semantic encoder is initialized with frozen Whisper-small weights and includes adapter modules, while the acoustic encoder is trainable from scratch.

**Comparison between XY-Tokenizer and FlexiCodec**    While the concurrent work XY-Tokenizer also targets low-frame-rate coding (12.5Hz) using ASR-augmented semantics, our method differs fundamentally in design philosophy. Beyond the distinction of dynamic versus fixed frame rates, we highlight two key structural differences:

- **Supervision and Scalability:** XY-Tokenizer relies on an explicit ASR loss, utilizing paired text transcripts and a frozen 0.5B LLM decoder during training. In contrast, FlexiCodec treats the ASR model solely as a frozen feature extractor and utilizes a feature reconstruction objective. This allows FlexiCodec to be trained entirely on unlabeled audio data, offering greater scalability and reduced training resource requirements. However, the text supervision from XY-Tokenizer may offer higher upper bounds, with an additional benefit of being aligned with an LLM's semantic space.

- **Disentanglement Strategy:** FlexiCodec achieves disentanglement via a hierarchical residual structure: the first quantizer layer (RVQ-1) explicitly captures semantics, while subsequent layers model the acoustic residuals. This flexible hierarchy allows downstream models to selectively utilize only the semantic tokens (RVQ-1) or the full stack for acoustic synthesis. By contrast, XY-Tokenizer employs a concatenative scheme without this hierarchical residual decoupling, which requires downstream models to processing the full set of tokens rather than offering a flexible, separable semantic layer.

We infer each baseline system from its official checkpoint. Each input audio has been resampled to match the codec's input frame rate. We also note that Mimi, XY-Tokenizer and SNAC do not support decoding solely from RVQ-1 tokens, so their RVQ-1 semantic tests are left blank in Table 5.

# H    MORE INFORMATION OF FLEXICODEC

**Codec Encoder and Decoder**    The acoustic codec encoder in FlexiCodec follows DAC, which is also used by DualCodec. It has a sequential series of 1D convolutional layers. It begins with an initial convolution with a kernel size of 7, followed by a set of residual convolutional blocks. Each block contains two dilated convolutions and a skip-connection, followed by a strided down-sampling layer. As the signal is down-sampled, the number of channels is doubled. A final 1D convolution sets the dimensionality of the output features. The decoder mirrors this architecture, using transposed convolutions in place of strided convolutions to upsample the signal, and reconstructs the final audio waveform. FlexiCodec has a codec encoder of 15M, and a codec decoder of 35M.

**Transformer Configurations**    FlexiCodec has a 20M parameter transformer in each frame merging module, each has 6 layers, 512 intermediate dimensions, 2048 FFN dimension, and 8 attention heads. The frame unmerging transformer has 100M parameters, with 32 layers, 2048 FFN dimension, and 8 attention heads. The transformers have bidirectional attentions with rotary postional embeddings. We also find that delaying the parameter updates of these transformers can help training stability, especially the unstable RVQ losses. We recommend bypassing them (setting them as identity functions) in the initial training steps, e.g., in the first 10% training steps.

**Loss Weights**    We use a weight of 15.0 for the multi-scale spectrogram loss, 2.0 for the GAN feature matching loss, 1.0 for the adversarial loss, 15.0 for the semantic feature alignment loss $L_{\text{feat}}$, 1.0 and 0.25 for the RVQ codebook and commitment losses, respectively.

The parameter breakdown of FlexiCodec is shown in Table 14. Additionally, the discriminators have 54M parameters in total, but is not used during inference.

Table 14: Number of parameters breakdown in FlexiCodec

| Component | Parameters |
|---|---|
| ASR model | 234M (frozen) |
| Codec encoder | 15M |
| Merging transformer (semantic stream) | 20M |
| Merging transformer (acoustic stream) | 20M |
| FSQ (semantic stream) | 26M (w/ ConvNeXt module) |
| RVQ (acoustic stream) | 0M |
| Unmerging transformer | 100M |
| Codec decoder | 35M |
| Total | 216M (trainable), 450M (whole) |

## I ASR PROBING TASK DETAILS

Following XARES (Zhang et al., 2025c) benchmark, the ASR probing task adds an MLP adaptor before a frozen Qwen2.5-0.5B model. The LM sequence is a concatenation of the adaptor-transformed speech feature, a separator token, and the ground truth text transcription. The model is trained using 1 Nvidia V100 GPU for 10 epochs, a batch size of 16, a cosine decay learning rate of initially $10^{-3}$, and decays to $10^{-4}$ over 10 epochs. During inference, a greedy auto-regressive sampling is used to predict the text tokens given the encoded speech features. The feature from FlexiCodec is obtained by using the FSQ continuous features of RVQ-1 tokens. Each frame is repeated by its "length" times to become 12.5Hz sequence. This is to provide the frame duration information to the ASR, and because XARES only accepts fixed-frame-rate token sequence.

## J LIMITATION AND FUTURE WORK

While FlexiCodec demonstrates strong performance in audio compression, semantic preservation, and flexible frame rate, there remain some limitations worth addressing in future research.

• **Multilingual and streaming support.** First, our experiments and model training have been conducted on English speech data. The current framework has not been evaluated or adapted for multilingual or code-switched speech, which presents unique challenges in semantic compression and phonetic variability. Extending FlexiCodec to handle multilingual corpora is an important direction. Second, FlexiCodec presently operates in a full-utterance mode and does not support streaming. We have recently seen unofficial adaptations of Sensevoice-Small ASR model for streaming, so adapting FlexiCodec for streaming is technically feasible. A combination of FlexiCodec's low frame rate and streaming capability will enable more efficient downstream tasks.

• **Improving interpretability.** While we have attempted to interpret the frame merging patterns by aligning with phonemes, it would be interesting to investigate if there is a mapping between phonemes and the semantic token index as well as its frame length. We plan to investigate this in a rule-based speech editing task.

• **Audio LLM.** We also plan to investigate the integration of FlexiCodec within unified multi-modal understanding and generation frameworks, such as audio LLMs. Most current audio LLMs use different features for understanding and generation. We will investigate using FlexiCodec tokens for joint semantic understanding and acoustic generation.

• **Improvements inspired by syllable unit discovery methods.** Although FlexiCodec was not explicitly designed to model syllables, our visualization analysis (Figure 4 and 5) suggests that the learned dynamic boundaries align closely with syllabic structures. A promising future direction is to explicitly incorporate syllable-level knowledge to guide the merging process. For instance, future work can explore using specialized syllable extractors like Sylber (Cho et al., 2024a) to replace

or augment the current semantic guidance, potentially achieving an even better trade-off between semantic abstraction and acoustic reconstruction.

• **Improving the dynamic frame rate algorithm.** In this work, we employed a simple and efficient greedy left-to-right algorithm for the dynamic frame merging process, based on the knowledge of pre-trained ASR models. This merging process contains no learnable parameters. While it is empirically effective, more sophisticated strategies could be investigated in future work. A particularly promising direction is to replace the current heuristic-based algorithm with a **learnable, end-to-end merging module**. Such a module could be implemented as a Transformer network that considers the sequence of ASR/syllabic/phonetic/acoustic features and outputs a sequence of merge decisions. By making this module differentiable (e.g., using Gumbel-Softmax or straight-through estimators for discrete decisions), its parameters could be optimized directly via backpropagation from the final reconstruction loss. The gradients from the decoder would act as a global signal, teaching the merging module to make decisions that are not just locally coherent but are **globally optimal** for minimizing reconstruction error. This approach offers several potential advantages:

- **Task-Optimized Segmentation:** Instead of relying on a proxy metric like cosine similarity, the model would learn a merging strategy tailored specifically to the end-goal of semantic reconstruction or high-fidelity audio reconstruction. It could learn, for instance, that preserving certain transient phonetic features is more critical for the decoder than preserving long, homogenous vowel sounds, and adjust its merging decisions accordingly.
- **Beyond Heuristics:** A learnable module could capture complex relationships in the data that our similarity threshold cannot. It could become sensitive to prosodic contours, speaker-specific timing, and other subtle cues that influence perceptual quality.

Developing such a learnable dynamic rate mechanism could potentially achieve an even more powerful trade-off between compression and quality.

## K    DECLARATION OF THE USE OF LLMS

During the preparation of this manuscript, we utilized large language models (LLMs) to assist with improving the grammar and readability of the text. The LLMs were used solely as a writing and editing tool and did not contribute to the generation of any original ideas, methods, or conclusions presented in this paper.

## L    TTS SUBJECTIVE EVALUATION INSTRUCTIONS

The instructions for our TTS subjective evaluation are shown in Table 15 and 16.

Table 15: Evaluation instructions for Naturalness Mean Opinion Score (NMOS)

| Score | Description |
|---|---|
| 1: Unacceptable | The voice lacks authenticity and sounds 100% robotic and mechanical. |
| 2: Poor | The voice sounds scripted and not human. |
| 3: Acceptable | The voice sounds human like, but stiff and scripted at the same time. |
| 4: Good | Mostly natural, with some minor scripted moments. |
| 5: Excellent | The voice feels genuinely human. |
| *Instruction:* Does the voice sound robotic or more human-like? | |

Table 16: Evaluation instructions for audio Quality Mean Opinion Score (QMOS)

| Score |
| --- |
| 1: Unacceptable |
| 2: Poor |
| 3: Acceptable |
| 4: Good |
| 5: Excellent |
| *Instruction:* Does it sound like a high-quality recording, without any technical difficulties (e.g., echoes, static, interference)? |

