# OpenReview forum: "FlexiCodec: A Dynamic Neural Audio Codec for Low Frame Rates"
_ICLR.cc/2026/Conference — ICLR 2026 Poster_

### Official Review · Reviewer_U5qj · 2025-10-25

**Soundness:** 4
**Presentation:** 3
**Contribution:** 3
**Rating:** 8
**Confidence:** 4

**Summary:**

They proposes a new codec with the following innovations:
1. frame merging based on ASR feature similarity, which leads to very low framerate codes
2. ASR feature condition, for frame merging, and also for RVQ-1

Extensive experiments show that FlexiCodec achieves good semantic and acoustic information preservation, as well as good performance in downstream NCLM-based TTS task.

**Strengths:**

1. Good angle: enhancing semantic info and representation, as well as low framerate compression are very important tasks
2. Good approach: leverage ASR features is probably the most straightforward approach to enhancing semantic info. Leveraging ASR features similarity for merging has a byproduct which is that we could adjust the compression rate based on the latency requirement
3. Very extensive and rigorous experiments: the authors not only did ablation studies on most design choices, standard reconstruction-based eval on WER, and acoustic info, but also shown that the alignment between dynamic merging and phonemes, as well as performance on downstream tasks TTS and audio understanding

**Weaknesses:**

1. XYcodec also uses ASR features, although in a slightly different way - the ASR module is finetuned during training with transcript - which will likely require more resources. Although in table 5 XYcodec is compared, I'd like to see more discussion on comparing XYcodec and FlexiCodec

2. the use of ASR features put FlexiCodec in a different category than most codec models, because it requires supervised signal. It would enhance the paper if the author can show that English-ASR trained model can also do well on unseen languages.

**Questions:**

Why do we use FSQ for RVQ-1, as opposed to VQ?

---

> ### Author Response · Authors · 2025-11-20
> **Response to reviewer U5qj (Part 1/2)**
>
> Dear Reviewer U5qj,
>
> Many thanks for your valuable comments and questions, which have helped us improve our work! We’ve made modifications in order to address your concerns, and the revised manuscript has been uploaded.
>
> ---
>
> >  [Q1] Why do we use FSQ for RVQ-1, as opposed to VQ?
>
> [A1] We used FSQ as RVQ-1 because it shows better results empirically. We ablated this design choice in Appendix D.1, row D1. As shown in the table below, which compares our baseline model using FSQ with a variant that uses a standard VQ for the first quantizer:
>
> | Label | Ablated Item          | WER (RVQ1)↓ | WER (RVQ1:8)↓ |  PESQ↑   |   SIM↑   |
> | :---- | :-------------------- | :---------: | :-----------: | :------: | :------: |
> | A1    | Baseline (FSQ)        |  **4.15**   |   **2.53**    | **2.76** | **0.71** |
> | D1    | Use VQ instead of FSQ |    4.43     |     2.66      |   2.74   |   0.68   |
>
> The results indicate that using FSQ is beneficial. Replacing FSQ with VQ (Row D1) leads to a degradation in performance across key metrics. To explain for the better performance of FSQ, we compared the token utilization rate during testing:
>
> | Model | RVQ-1 token utilization |
> | :--- | :---: |
> | Baseline (FSQ) | 18035 / 32768 |
> | Use VQ instead of FSQ | 11770 / 32768 |
>
> As shown above, the FSQ-based model utilizes a significantly larger portion of its codebook (18,035 unique tokens) compared to the VQ-based model (11,770 unique tokens) out of a total possible 32,768. The higher token utilization (avoiding codebook collapse) is a feature of FSQ highlighted in its paper. Another difference is that FSQ does not require auxiliary losses in VQ, including codebook learning loss and commitment loss, avoiding the process of finding optimial loss weights.
>
> Although successfully applied to the semantic VQ, we had not applied FSQ to the acoustic RVQ because FSQ is a single-layer quantization. As another reviewer suggested, it is possible to use a more advanced rFSQ scheme for acoustic quantization. We think this is a promising direction to improve our work.
>
>
>
> >  [Weakness, 1] XYcodec also uses ASR features, although in a slightly different way - the ASR module is finetuned during training with transcript - which will likely require more resources. Although in table 5 XYcodec is compared, I'd like to see more discussion on comparing XYcodec and FlexiCodec.
>
> [A2] Sure. We agree that a detailed comparison with the concurrent work XY-Tokenizer is crucial for contextualizing our contributions. To provide this, we have added new paragraphs in our **Appendix G.2** titled "Comparison between XY-Tokenizer and FlexiCodec." In this section, we move beyond the empirical results in Table 5 to discuss the differences in design philosophy and architecture. For the reviewer's convenience, we have pasted the full text of this new paragraph below:
>
> **Comparison between XY-Tokenizer and FlexiCodec:** While the concurrent work XY-Tokenizer also targets low-frame-rate coding (12.5Hz) using ASR-augmented semantics, our method differs fundamentally in design philosophy. Beyond the distinction of **dynamic versus fixed frame rates**, we highlight two key structural differences:
>
> - **Supervision and Scalability**: XY-Tokenizer relies on an explicit ASR loss, utilizing paired text transcripts and a frozen 0.5B LLM decoder during training. In contrast, FlexiCodec treats the ASR model solely as a frozen feature extractor and utilizes a feature reconstruction objective. This allows FlexiCodec to be trained entirely on unlabeled audio data, offering greater scalability and reduced training resource requirements. However, the text supervision from XY-Tokenizer may offer higher upper bounds, with an additional benefit of being aligned with an LLM's semantic space.
>
> - **Disentanglement Strategy**: FlexiCodec achieves disentanglement via a hierarchical residual structure: the first quantizer layer (RVQ-1) explicitly captures semantics, while subsequent layers model the acoustic residuals. This flexible hierarchy allows downstream models to selectively utilize only the semantic tokens (RVQ-1) or the full stack for acoustic synthesis. By contrast, XY-Tokenizer employs a concatenative scheme without this hierarchical residual decoupling, which requires downstream models to processing the full set of tokens rather than offering a flexible, separable semantic layer.

---

> > ### Author Response · Authors · 2025-11-20
> > **Response to reviewer U5qj (Part 2/2)**
> >
> > >  [Weakness,2] The use of ASR features put FlexiCodec in a different category than most codec models, because it requires supervised signal. It would enhance the paper if the author can show that English-ASR trained model can also do well on unseen languages.
> >
> > [A3] We thank the reviewer for raising this critical point about generalizability to unseen languages. To address this directly, we have conducted a new set of experiments and added them to **Appendix F**. In this section, we evaluate FlexiCodec on the **Emilia dataset**, which consists of in-the-wild, spontaneous, and conversational speech — a significant shift from the read audiobooks of LibriLight. Our experiment contains both English testing and **multilingual** testing. Because of the large sizes of the table, we're unable to attach them here, and we kindly ask the reviewer to examine the revised manuscript. Our analysis of the new evaluation is as follows:
> >
> > 1. **Acoustic Cross-Lingual Generalization**: We tested our English-trained model on five unseen languages (ZH, KO, JA, FR, DE). While the semantic-only layer (RVQ1) expectedly struggles, the full acoustic reconstruction (RVQ1:8) demonstrates remarkable robustness. Because of the large table size, we're unable to attach it here, and we kindly ask the reviewer to examine our revised manuscript. As shown in **Table 11** (Appendix F), our model achieves WER(RVQ1:8) and SIM scores that are competitive with baselines specifically trained on multilingual data. This indicates that the acoustic modeling of FlexiCodec generalizes well across languages, even without explicit multilingual training.
> >
> > 2. **Efficient Adaptation via Fine-Tuning**: To assess the effort required for adaptation, we fine-tuned our model in an unseen language: Chinese. The results in Table 11 ("FlexiCodec-ZH_tune") are compelling: (1) Dramatic Target-Language Improvement: Performance on Chinese improved significantly, with the WER(RVQ1) score dropping from 100 to just 8.5, outperforming several in-domain baselines. It also confirms that our dynamic frame rate works outside of the English language. (2) Positive Zero-Shot Transfer: Fine-tuning in Chinese also improved zero-shot performance on other unseen languages like Japanese and Korean.
> >
> > We believe these results demonstrate that (1) FlexiCodec possesses acoustic cross-lingual generalization capabilities, and (2) its semantic preservation can be efficiently adapted to new languages, achieving strong performance without requiring a complete retraining or a massive language-specific ASR model from scratch.
> >
> > ---
> >
> > Thank you again for your reviewing efforts. We’re happy to address any more comments or concerns you may have.

---

> > > ### Comment · Reviewer_U5qj · 2025-11-22
> > >
> > > Thanks for the reply. I'm very happy with the paper as is and therefore further raised my score

---

> > > > ### Author Response · Authors · 2025-11-23
> > > > **Thanks to reviewer U5qj**
> > > >
> > > > Dear reviewer U5qj,
> > > >
> > > > We are incredibly grateful for your positive feedback and for further increasing your ratings of our paper. Your encouragement means a great deal to us.
> > > >
> > > > Thank you again for your time, expertise, and for helping us improve this paper. We’re happy to address any more comments you may have.

---

### Official Review · Reviewer_f4v8 · 2025-10-31

**Soundness:** 3
**Presentation:** 3
**Contribution:** 3
**Rating:** 6
**Confidence:** 4

**Summary:**

This paper introduces FlexiCodec, a novel dynamic-rate neural audio codec that successfully addresses the critical problem of semantic information loss at very low frame rates. Its core ideas are innovative and well-executed, specifically the use of a pre-trained ASR model to guide adaptive frame merging and the implementation of controllable frame rates. Supported by extensive and rigorous experiments against strong baselines, the work presents a compelling case for its effectiveness and makes a significant contribution to the field of low-bitrate speech tokenization for language models.

**Strengths:**

1. The core contribution of a dynamic and controllable frame rate is highly novel and effectively solves a well-motivated problem. The adaptive allocation of temporal resolution based on phonetic complexity is an elegant and powerful approach to preserving semantics.

2. The architecture is thoughtfully designed, cleverly using a pre-trained ASR model for the dual purpose of providing semantic features and guiding the frame merging process. The inclusion of Transformer modules for refinement is also a crucial detail for ensuring high quality.

3. The evaluation is exceptionally thorough and convincing. The authors compare against fairly retrained baselines, conduct extensive ablation studies that validate key design choices, and demonstrate practical utility in downstream TTS and audio understanding tasks.

**Weaknesses:**

1. The model's evaluation is confined to English, and its reliance on a language-specific ASR model raises concerns about its generalizability to multilingual settings without significant additional effort or resources.

2. A more detailed analysis of the codec's own computational overhead and latency during encoding/decoding would be beneficial to fully assess its efficiency.

3. The frame merging process relies on a simple, greedy left-to-right algorithm. While empirically effective, it may not be globally optimal, and a discussion of more sophisticated segmentation strategies could have strengthened the work..

**Questions:**

N/A

---

> ### Author Response · Authors · 2025-11-20
> **Response to reviewer f4v8 (Part 1/2)**
>
> Dear Reviewer f4v8,
>
> Many thanks for your valuable comments and questions, which have helped us improve our work. We address your points as follows. We’ve made modifications in order to address your concerns, and the revised manuscript has been uploaded.
>
> ---
>
> >  [Q1] The model's evaluation is confined to English, and its reliance on a language-specific ASR model raises concerns about its generalizability to multilingual settings without significant additional effort or resources.
>
> [A1] We thank the reviewer for raising this critical point. We agree that demonstrating multilingual generalizability is important. To address this, we have conducted a comprehensive new set of experiments in **Appendix F**, evaluating FlexiCodec on the **multilingual and out-of-domain Emilia dataset**. Due to the limited space, we could not attach the table here, and we hope the reviewer could examine our revised manuscript. Our findings are as follows:
>
> 1. **Acoustic Cross-Lingual Generalization**: We tested our English-trained model on five unseen languages (ZH, KO, JA, FR, DE). While the semantic-only layer (RVQ1) expectedly struggles, the full acoustic reconstruction (RVQ1:8) demonstrates remarkable robustness. Because of the large table size, we're unable to attach it here, and we kindly ask the reviewer to examine our revised manuscript. As shown in **Table 11** (Appendix F), our model achieves WER(RVQ1:8) and SIM scores that are competitive with baselines specifically trained on multilingual data. This indicates that the acoustic modeling of FlexiCodec generalizes well across languages, even without explicit multilingual training.
>
> 2. **Efficient Adaptation via Fine-Tuning**: To assess the effort required for adaptation, we fine-tuned our model in an unseen language: Chinese. The results in Table 11 ("FlexiCodec-ZH_tune") are compelling: (1) Dramatic Target-Language Improvement: Performance on Chinese improved significantly, with the WER(RVQ1) score dropping from 100 to just 8.5, outperforming several in-domain baselines. It also confirms that our dynamic frame rate works outside of the English language. (2) Positive Zero-Shot Transfer: Fine-tuning in Chinese also improved zero-shot performance on other unseen languages like Japanese and Korean.
>
> We believe these results demonstrate that (1) FlexiCodec possesses acoustic cross-lingual generalization capabilities, and (2) its semantic preservation can be efficiently adapted to new languages, achieving strong performance without requiring a complete retraining or a massive language-specific ASR model from scratch.
>
> >  [Q2] A more detailed analysis of the codec's own computational overhead and latency during encoding/decoding would be beneficial to fully assess its efficiency.
>
> [A2] We agree that evaluating FlexiCodec's inference speed and consider its model size is critical for assessing the model's practical usability. Therefore, we have added **Appendix E** in our revision, that presents the efficiency evaluation of FlexiCodec. Our analysis shows that, while FlexiCodec is not the most efficient among the evaluated systems, it still demonstrates very acceptable efficiency. Its encoding and decoding RTF remains constant at 0.018 (encode) and 0.006 (decode) across different frame rate options (12.5Hz, 8.3Hz, and 6.25Hz). The low RTF values indicate that FlexiCodec only adds a very small overhead to downstream tasks like TTS, whose RTFs are usually more than 0.1. This positions FlexiCodec as a practical and scalable solution for real-world applications.
>
>
> | Model                   | Encode Params | Encode RTF↓ | Decode Params | Decode RTF↓ |
> |-------------------------|--------------|------------|--------------|------------|
> | DAC-official-75Hz       | 22M          | 0.004      | 52M          | 0.002      |
> | Encodec-75Hz            | 7.4M         | 0.004      | 7.4M         | 0.004      |
> | SpeechTokenizer-50Hz    | 68M          | 0.005      | 35M          | 0.007      |
> | WavTokenizer-75Hz       | 8.8M         | 0.005      | 62M          | 0.002      |
> | Mimi-12.5Hz             | 38M          | 0.003      | 40M          | 0.002      |
> | SemantiCodec-50Hz       | 725M         | 0.100      | 196M         | 0.650      |
> | SemantiCodec-25Hz       | 1660M        | 0.120      | 218M         | 0.643      |
> | XYTokenizer-12.5Hz      | 306M         | 0.022      | 214M         | 0.005      |
> | DualCodec-12.5Hz        | 622M         | 0.019      | 53M          | 0.002      |
> | FlexiCodec @12.5Hz       | 289M         | 0.018      | 161M         | 0.006      |
> | FlexiCodec @8.3Hz        | 289M         | 0.018      | 161M         | 0.006      |
> | FlexiCodec @6.25Hz       | 289M         | 0.018      | 161M         | 0.006      |

---

> > ### Author Response · Authors · 2025-11-20
> > **Response to reviewer f4v8 (Part 2/2)**
> >
> > >  [Q3] The frame merging process relies on a simple, greedy left-to-right algorithm. While empirically effective, it may not be globally optimal, and a discussion of more sophisticated segmentation strategies could have strengthened the work.
> >
> > [A3] Thank you for your insightful suggestion. We have added a new discussion to our Limitations and Future Work section (**Appendix J)** under "Improving the dynamic frame rate algorithm." We discuss a promising future direction: replacing the current heuristic-based algorithm with a learnable, end-to-end merging module. For clarity, we have pasted the full text of the new paragraph below:
> >
> > - In this work, we employed a simple and efficient greedy left-to-right algorithm for the dynamic frame merging process, based on the knowledge of pre-trained ASR models. This merging process contains no learnable parameters. While it is empirically effective, more sophisticated strategies could be investigated in future work. A particularly promising direction is to replace the current heuristic-based algorithm with **a learnable, end-to-end merging module**.
> >
> > - Such a module could be implemented as a Transformer network that considers the sequence of ASR/syllabic/phonetic/acoustic features and outputs a sequence of merge decisions. By making this module differentiable (e.g., using Gumbel-Softmax or straight-through estimators for discrete decisions), its parameters could be optimized directly via backpropagation from the final reconstruction loss. The gradients from the decoder would act as a global signal, teaching the merging module to make decisions that are not just locally coherent but are globally optimal for minimizing reconstruction error. This approach offers several potential advantages:
> >
> >     - Task-Optimized Segmentation: Instead of relying on a proxy metric like cosine similarity, the model would learn a merging strategy tailored specifically to the end-goal of semantic reconstruction or high-fidelity audio reconstruction. It could learn, for instance, that preserving certain transient phonetic features is more critical for the decoder than preserving long, homogenous vowel sounds, and adjust its merging decisions accordingly.
> >
> >     - Beyond Heuristics: A learnable module could capture complex relationships in the data that our similarity threshold cannot. It could become sensitive to prosodic contours, speaker-specific timing, and other subtle cues that influence perceptual quality.
> >
> >
> > ---
> >
> > Thank you again for your insightful and professional comment, please let us know if there are any other changes that would improve the paper in your view.

---

### Official Review · Reviewer_VFuy · 2025-10-31

**Soundness:** 2
**Presentation:** 2
**Contribution:** 2
**Rating:** 4
**Confidence:** 3

**Summary:**

This paper employs a dynamic neural audio codec, which adaptively merges segments of varying lengths according to different speaking rates to achieve efficient information compression.

**Strengths:**

The use of ASR features as auxiliary information is well-motivated and enables more effective compression. This design allows the model to merge segments corresponding to the same phoneme, thereby shortening the overall sequence length while retaining essential information.

**Weaknesses:**

I am not entirely certain about my understanding of the implementation. Does each token need to maintain an additional length information field? If so, should this length information also be considered part of the compressed representation, since it seems necessary for accurate audio reconstruction? Clarifying this design choice and its impact on bitrate or compression ratio would strengthen the paper.

**Questions:**

How does the proposed method handle fast speech scenarios, where phonemes or syllables occur at very high rates? Can the model still achieve low compression rates while maintaining intelligibility and reconstruction quality in such cases? A discussion or experiment addressing this would be valuable.

---

> ### Author Response · Authors · 2025-11-20
> **Response to reviewer VFuy (Part 1/2)**
>
> Dear Reviewer VFuy,
>
> Many thanks for your valuable comments and questions, which have helped us improve our work. We address your points as follows. We’ve made modifications in order to address your concerns, and the revised manuscript has been uploaded.
>
> >  [Weakness,1] Does each token need to maintain an additional length information field? If so, should this length information also be considered part of the compressed representation?
>
> [A] Yes. The decoder requires the duration (length) of each frame to reconstruct the audio faithfully. In the following, we first explain more about the additional length information field, confirm its bitrate overhead, and detail our added clarification in the manuscript to address your concerns.
>
> - **Included in Compressed Representation**: Yes, the length information is an integral part of the compressed representation. Each token index must be paired with its duration, both are stored/transmitted together in audio compression task. In a downstream TTS task, however, the length information of each token is predicted by a separate LM head.
>
> - **Accounted for** **in** **Bitrate**: We confirm that our reported bitrates in Table 5 already include this overhead. The reported bitrates of FlexiCodec already account for both the quantized tokens and their corresponding length attributes. Specifically, we allocate 3 bits per frame for length information (The maximum frame length is limited to 8, $\log_2(8)=3$ bits).
>
> - **Clarification in Revised Manuscript**: We apologize that this was not explicitly detailed in the initial draft. We have now added a footnote to Page 9 to prevent any ambiguity: “Storing each of FlexiCodec’s frame length attributes requires $\log_2(8)=3$ bits. Consequently, our bitrate calculations in Table 5 already include an additional $3 \times $  frame rate bits per second to account for this overhead.”

---

> > ### Author Response · Authors · 2025-11-20
> > **Response to reviewer VFuy (Part 2/2)**
> >
> > >  [Q1] How does the proposed method handle fast speech scenarios, where phonemes or syllables occur at very high rates? Can the model still achieve low compression rates while maintaining intelligibility and reconstruction quality in such cases?
> >
> > [A1] Thanks for the insightful question. I think this points to a core strength of our method, that FlexiCodec is inherently well-suited for fast speech because its frame rate is dynamic, not fixed.
> >
> > - In fast speech, phonetic units are shorter and change more rapidly. Our ASR-guided merging algorithm detects these frequent semantic shifts and consequently performs less merging. This adaptively increases the local frame rate in these dense regions to capture all necessary phonetic details. **This behavior is demonstrated in Figure 4, which shows that when a person speaks very fast (high phoneme rate), our codec uses higher output frame rate**; in fact, it shows a linear correlation with the underlying phoneme rate. In the figure, the largest value in the x axis is 20 phonemes/s (a very fast speaking rate), and our codec outputs at nearly 12 token/s (12Hz). Cross-checking with Figure 3 shows that at 12.5Hz, our codec can achieve near-ground truth intelligibility and high reconstruction quality in this case.
> >
> > - Regarding the second part of your question—whether the model can still achieve a low average compression rate—the answer is yes. This is because fast speech segments, or conversations, are typically balanced by periods of slower speech, pauses, or silence within the same audio clip. Our model capitalizes on these information-sparse regions by merging frames aggressively, significantly lowering the frame rate there. For example, **Figure 5 in the Appendix A shows that the trailing silence region is frequently merged** **in** **different audios.**
> >
> > - Therefore, even if an utterance contains challenging bursts of fast speech where the local frame rate is high, the overall average frame rate for the entire utterance, or a pool of utterances, remains low. This allows FlexiCodec to maintain high intelligibility while still achieving efficient average compression rate (e.g., 6.25Hz or 8.3Hz). **We also provide real-world cases on our demo page, where samples 7, 8 and 9 are at fast speaking rates, and FlexiCodec achieves great intelligibility and reconstruction quality at each target frame rate.**
> >
> > - **We have strengthened our paper with out-of-domain testing to verify FlexiCodec's robustness.** We have added a new section, **Appendix F,** to comprehensively evaluate FlexiCodec on out-of-domain (OOD) and multilingual scenarios using the **Emilia dataset** (spontaneous, in-the-wild speech, contains more fast speaking rate samples). Our results confirm that FlexiCodec maintains high semantic preservation and acoustic quality in the English Emilia dataset, and can be adapted for an OOD language via finetuning.
> >
> >
> > ---
> >
> > Thank you again for helping us improve the paper and hope our response can resolve your concerns! **We would be very grateful if you would consider re-evaluating our work in light of our discussion and revision**. Please let us know if you have any further questions.

---

### Official Review · Reviewer_URnt · 2025-11-01

**Soundness:** 2
**Presentation:** 3
**Contribution:** 2
**Rating:** 2
**Confidence:** 4

**Summary:**

This paper introduces FlexiCodec, a neural audio codec designed to operate at very low and controllable frame rates. The core contribution is a dynamic frame rate mechanism that adaptively merges semantically similar frames, guided by features from a pre-trained ASR model. This is implemented within a dual-stream architecture (semantic and acoustic) that utilizes Transformer-based modules for merging and unmerging frames. The authors demonstrate that this approach improves the preservation of semantic information. The paper also validates FlexiCodec's effectiveness in a downstream TTS application, showing competitive audio quality with substantial inference speedups in the autoregressive stage.

**Strengths:**

1. The idea of using pre-trained ASR features to guide the merging of semantically similar frames is intuitive and well-executed. The results in Figure 3a, showing a dramatic improvement in RVQ-1 WER at 6.25Hz, strongly validate this approach's effectiveness in preserving semantic content on in-domain data.
2. The design allows for a flexible trade-off between semantic quality and sequence length at inference time by simply adjusting the similarity threshold. This is a highly practical feature for applications with varying computational constraints.
3. The authors conduct a comprehensive set of experiments for their chosen domain, including detailed ablations of the dynamic rate mechanism (Tables 3 & 4), comparisons with numerous existing codecs, and validation on two distinct downstream tasks (TTS and audio understanding).

**Weaknesses:**

1. The abstract claims, "We find that a major challenge for very low frame rate tokens is missing semantic information". I do not fully agree. Recent work on syllabic / dynamic units, specifically Sylber and SyllableLM (Cho et al., 2024, Baade et al., 2024), showed that at around 6-8 Hz you can still carry the linguistic sequence reasonably well, and what starts to go missing is acoustics/prosody/fine timing, not semantics. Figure 4 of the paper actually supports a syllable-like view: FlexiCodec emits about half the phoneme rate, i.e., around syllabic granularity. The authors should tone down this claim and provide a more nuanced motivation that acknowledges that while fixed downsampling can lose transient phonetic details, the main trade-off at very low rates is often acoustic fidelity vs. semantic representation.
2. The ASR encoder (Sense Voice-Small) was trained on 300k hours of data , and FlexiCodec itself is trained on Librilight-Large (54k hours of audiobooks). The primary evaluation is on LibriSpeech-test-clean, which is also an audiobook dataset. This creates a risk that the excellent WER results are due to the ASR features being highly specialized for clean, read English speech. The claims of superior semantic preservation would be far more compelling if they were supported by an evaluation on an out-of-domain (OOD) dataset, for instance, a corpus of spontaneous or conversational speech (Emilia). This would test whether the ASR-guided merging generalizes beyond the training domain.
3. The full FlexiCodec model has 216M trainable parameters, with the Frame Unmerging Module alone containing a 100M parameter Transformer. This is a substantial model. While the paper provides Real-Time Factor (RTF) for the downstream TTS task, it omits the RTF for the codec's own encoding and decoding process. This information is critical for assessing the model's practical usability. A model that is fast for downstream tasks but slow to encode/decode may have limited applications.
4. Table 3 says: removing dynamic frame rate at 6.25 Hz increases RVQ1 WER and probing WER. That’s good, but this ablation is entangled with (a) ASR features, (b) FSQ, (c) transformer smoothing. Right now we cannot tell whether: 1) ASR features alone at 6.25 Hz already close most of the gap; 2) FSQ is the key piece at low rate; 3) or dynamic merging is the actual differentiator. I would suggest providing a factorial ablation: 1) DualCodec-style SSL features, fixed 6.25 Hz; 2) ASR features, fixed 6.25 Hz; 3)ASR features, dynamic 6.25 Hz; 4) FSQ. That way, we can see the incremental lift of each choice.
5. The paper states: “To our best knowledge, it is the one of first neural audio codecs under 10Hz… and the first work to explore dynamic frame rate on low-frame-rate neural audio codecs.” But there are concurrent <= 10 Hz speech-token systems (TaDiCodec, TASTE), the authors mention them later, and several dynamic-rate codecs, though at higher base rates. This should be toned down.

**Questions:**

See Weaknesses

---

> ### Author Response · Authors · 2025-11-20
> **Response to reviewer URnt (Part 1/3)**
>
> Dear Reviewer URnt,
>
> Many thanks to your valuable comments and questions, which help us a lot to improve our work. We’ve made a number of modifications in order to address your concerns. Our revised manuscript has been uploaded, with the changes highlighted in red. Our response to your questions and suggestions are as follows.
>
> ---
>
> >  [Q1] The authors should tone down this claim ("We find that a major challenge for very low frame rate tokens is missing semantic information") and provide a more nuanced motivation that acknowledges that while fixed downsampling can lose transient phonetic details, the main trade-off at very low rates is often acoustic fidelity vs. Semantic representation.
>
> [A1] We thank the reviewer for this insightful feedback and for pointing us to the relevant literature on syllabic units. We have made the following revisions to address them:
>
> 1. **Revised Motivation**: We have revised the **Abstract** and **Introduction** to present a more nuanced motivation. We now frame the major problem around th trade-off between semantic preservation and acoustic fidelity at very low frame rates: "Existing codecs may be primarily limited by an insufficient decoupling of semantics from acoustics." We explicitly reference and discuss recent work on syllable-based unit extraction, including Sylber and SyllableLM (as SylBoost) in our motivation discussion.
>
> 2. **Extended Literature Review**: To make our review more complete, we have added Sylber to our related works (Section 2.2) on syllable unit extractors, expanding our original literature review on syllable extractors. The other work you mentioned, SyllebleLM (SylBoost), has been reviewed in our first draft.
>
> 3. **Acknowledged Syllabic Correlation and Future Work**: We appreciate the reviewer's observation that our model's dynamic units show a strong correlation with syllabic structures. This is an excellent point that opens up exciting avenues for future research. We have added a new paragraph to our **Appendix J (Future Work)** to explicitly discuss this. The new paragraph states: "Although FlexiCodec was not explicitly designed to model syllables, our visualization analysis (Figure 4 and 5) suggests that the learned dynamic boundaries align closely with syllabic structures. A promising future direction is to explicitly incorporate syllable-level knowledge to guide the merging process. For instance, future work can explore using specialized syllable extractors like Sylber to replace or augment the current semantic guidance, potentially achieving an even better trade-off between semantic abstraction and acoustic reconstruction."
>
>
>
> >  [Q2] The claims of superior semantic preservation would be far more compelling if they were supported by an evaluation on an out-of-domain (OOD) dataset, for instance, a corpus of spontaneous or conversational speech (Emilia). This would test whether the ASR-guided merging generalizes beyond the training domain.
>
> [A2] We agree that evaluating on out-of-domain (OOD) spontaneous speech is crucial to verify the robustness of our ASR-guided merging. To address this, we have added **Appendix F**, where we evaluate FlexiCodec on the **Emilia dataset**, which consists of in-the-wild, spontaneous, and conversational speech — a significant shift from the read audiobooks of LibriLight. Our experiment contains both English testing and **multilingual** testing. Because of the large sizes of the table, we're unable to attach them here, and we kindly ask the reviewer to examine the revised manuscript. Our analysis of the new evaluation is as follows:
>
> 1. **Robustness on Out-of-Domain English speech (Table 11):** Despite being trained on audiobooks, FlexiCodec generalizes effectively to spontaenous/conversational speech, especially demonstrating robust semantic preservation. On the English Emilia subset, the WER (RVQ1) scores for FlexiCodec@12.5Hz , 8.3Hz and 6.25Hz are 8.25%, 10.8%, and 12.9%. This performance is highly competitive with other state-of-the-art codecs (e.g., DualCodec-12.5Hz has 14.2%, WavTokenizer-75Hz has 10.7%), **confirming that FlexiCodec's superior semantic preservation is not overfitted to clean, audiobook speech, but handles spontaneous, out-of-domain speech robustly**. The acoustic results are robust to OOD data as well.
>
> 2. **Cross-Lingual Generalization (Table 12):** We further pushed the generalization limit by testing on non-English languages in Emilia, including Chinese (ZH), Korean (KO), Japanese (JA), French (FR) and German (DE). Results show that while the semantic token (RVQ1) performance drops as expected (since the model is English-trained), the full acoustic reconstruction (RVQ1:8) remains strong (e.g., comparable WER/SIM to multilingual-trained baselines). Furthermore, we show that fine-tuning on Chinese data (FlexiCodec-ZH_tune) yields dramatic improvements not just for Chinese, but also zero-shot gains for Japanese and Korean, demonstrating high transferability.

---

> ### Author Response · Authors · 2025-11-20
> **Response to reviewer URnt (Part 2/3)**
>
> >  [Q3] Omission of the RTF for the codec's own encoding and decoding process.
>
> [A3] We agree that evaluating FlexiCodec's inference speed and consider its model size is critical for assessing the model's practical usability, which was missing from our first draft. Therefore, we have added **Appendix E** in our revision, that presents the **efficiency evaluation of FlexiCodec.** Our analysis shows that, while FlexiCodec is not the most efficient among the evaluated systems, it still demonstrates very acceptable efficiency. Its encoding and decoding RTF remains constant at 0.018 (encode) and 0.006 (decode) across different frame rate options (12.5Hz, 8.3Hz, and 6.25Hz). The low RTF values indicate that FlexiCodec only adds a very small overhead to downstream tasks like TTS, whose RTFs are usually more than 0.1. This positions FlexiCodec as a practical and scalable solution for real-world applications. The efficiency evaluation result is attached below:
>
>
> | Model                   | Encode Params | Encode RTF↓ | Decode Params | Decode RTF↓ |
> |-------------------------|--------------|------------|--------------|------------|
> | DAC-official-75Hz       | 22M          | 0.004      | 52M          | 0.002      |
> | Encodec-75Hz            | 7.4M         | 0.004      | 7.4M         | 0.004      |
> | SpeechTokenizer-50Hz    | 68M          | 0.005      | 35M          | 0.007      |
> | WavTokenizer-75Hz       | 8.8M         | 0.005      | 62M          | 0.002      |
> | Mimi-12.5Hz             | 38M          | 0.003      | 40M          | 0.002      |
> | SemantiCodec-50Hz       | 725M         | 0.100      | 196M         | 0.650      |
> | SemantiCodec-25Hz       | 1660M        | 0.120      | 218M         | 0.643      |
> | XYTokenizer-12.5Hz      | 306M         | 0.022      | 214M         | 0.005      |
> | DualCodec-12.5Hz        | 622M         | 0.019      | 53M          | 0.002      |
> | FlexiCodec @12.5Hz       | 289M         | 0.018      | 161M         | 0.006      |
> | FlexiCodec @8.3Hz        | 289M         | 0.018      | 161M         | 0.006      |
> | FlexiCodec @6.25Hz       | 289M         | 0.018      | 161M         | 0.006      |

---

> > ### Author Response · Authors · 2025-11-20
> > **Response to reviewer URnt (Part 3/3)**
> >
> > >  [Q4] Table 3 says: removing dynamic frame rate at 6.25 Hz increases RVQ1 WER and probing WER. That’s good, but this ablation is entangled with (a) ASR features, (b) FSQ, (c) transformer smoothing. Right now we cannot tell whether: 1) ASR features alone at 6.25 Hz already close most of the gap; 2) FSQ is the key piece at low rate; 3) or dynamic merging is the actual differentiator. I would suggest providing a factorial ablation: 1) DualCodec-style SSL features, fixed 6.25 Hz; 2) ASR features, fixed 6.25 Hz; 3)ASR features, dynamic 6.25 Hz; 4) FSQ. That way, we can see the incremental lift of each choice.
> >
> > [A4] Thank you for this constructive suggestion. We first wish to clarify the methodology of our original ablation, and then present the factorial analysis you suggested.
> >
> > - Clarification on Original Ablation (Table 3). We respectfully clarify that our original ablation in Table 3 ("removing dynamic frame rate") was not entangled with other factors. As detailed in L547, we isolated the dynamic rate by modifying the codec encoder strides to output a static 6.25Hz while maintaining the same (a) ASR features, (b) FSQ, and (c) Transformer modules (preserving total parameter count). Thus, Table 3 measured the impact of the dynamic mechanism alone.
> >
> > - **New** **Factorial** **Analysis (Table 9)**. We agree that explicitly decoupling every component provides a clearer view of the incremental gains. We have added a Factorial Analysis section in **Appendix D.2 (Table 9)**, comparing six configurations (F1–F6). We note that 5 of the 6 data points in the table have occured previously in the manuscript, so the table mainly provides a factorial view of the data. We have also labeled out the source of the data. Our analysis is as follows.
> >
> >
> > 1. **Changing from** **SSL** **to** **ASR** **feature is the primary driver for semantic preservation at a low frame rate.** As shown by comparing F1 and F2, replacing DualCodec-style SSL features (F1) with SenseVoice ASR features (F2) at a fixed 6.25Hz reduces WER dramatically from 31.5 to 5.99. This confirms that using a semantic-rich ASR feature is the primary driver for FlexiCodec's better semantic preservation.
> >
> > 2. **The dynamic rate mechanism is key to further** **improve** **semantic preservation.** Comparing the full fixed-rate model (F5) to the dynamic model (F6), the WER improves from 5.22 to 4.15. This ~20% relative improvement confirms that even with powerful ASR features, a fixed grid is suboptimal; our dynamic boundary detection captures semantic transitions that fixed striding misses.
> >
> > 3. **Transformers & FSQ provide incremental benefits** (F2 to F3, F4 to F6). Adding the Merging/Unmerging Transformers (F2 to F3) improves WER from 5.99 to 5.40. Note that the Transformer modules add 140M parameters to the model. Finally, FSQ (F4 to F6) offers incremental optimization to obtain our final FlexiCodec model, lowering WER from 4.43 to 4.15.
> >
> >
> > | Label | Semantic feat  | Dynamic rate? | Merging&Unmerging transformer? | FSQ? | WER(RVQ1) | Occurrence of data in manuscript          |
> > | :---- | :------------- | :-----------: | :----------------------------: | :--: | :-------: | :---------------------------------------- |
> > | F1    | w2v-bert-2 SSL |       ✗       |               ✗                |  ✗   |   31.5    | Figure 3 (the reproduced DualCodec at 6.25Hz) |
> > | F2    | SenseVoice ASR |       ✗       |               ✗                |  ✗   |   5.99    | Section 4.2 (briefly mentioned in text)   |
> > | F3    | SenseVoice ASR |       ✗       |               ✓                |  ✗   |   5.40    | -                                         |
> > | F4    | SenseVoice ASR |       ✓       |               ✓                |  ✗   |   4.43    | Table 8 (Ablation D1)                     |
> > | F5    | SenseVoice ASR |       ✗       |               ✓                |  ✓   |   5.22    | Table 3                                   |
> > | F6    | SenseVoice ASR |       ✓       |               ✓                |  ✓   | **4.15**  | Table 3&5                                 |
> >
> > >  [Q5] The paper states: “To our best knowledge, it is the one of first neural audio codecs under 10Hz… and the first work to explore dynamic frame rate on low-frame-rate neural audio codecs.” But there are concurrent <= 10 Hz speech-token systems (TaDiCodec, TASTE), the authors mention them later, and several dynamic-rate codecs, though at higher base rates. This should be toned down.
> >
> > [A5] Thank you for the constructive feedback. We have made our claim more appropriate in our revised manuscript (Page 2).
> >
> > ---
> >
> > Thank you again for helping us improve the paper and hope our response can resolve your concerns! **We would be very grateful if you would consider re-evaluating our work in light of these changes**. Please let us know if you have any further questions.

---

### Official Review · Reviewer_TQuw · 2025-11-01

**Soundness:** 3
**Presentation:** 3
**Contribution:** 3
**Rating:** 8
**Confidence:** 3

**Summary:**

This paper introduces FlexiCodec, a dynamic neural audio codec targeting particularly low frame rates (<12.5Hz) using a variable frame rate instead of a fixed frame rate, providing improvements particularly in semantic information preservation and acoustic quality.
It uses an ASR-feature-assisted dual stream architecture to allocate more frames for complex audio segments and fewer to  information-sparse subsequences as in silences or long vowels by adaptively merging adjacent frames.
Experiments show FlexiCodec significantly outperforms baselines in semantic preservation at 6.25Hz and 8.3Hz, while also supporting variable inference time frame rates (from 3 to 12.5Hz) to control potential trade-offs, and strong performance in downstream TTS tasks.

**Strengths:**

The paper presents strong empirical results and practical utility, with appropriate comparisons by e.g. retraining very recent work for lower frame rates. Previous work has primarily focused on lowering frame rates to 12.5Hz but not below.
- A core contribution is the dynamic frame rate mechanism, which (expectedly) primarily improves semantic preservation rather than acoustic representations
  - Experiments convincingly demonstrate that this dynamic approach significantly improves semantic preservation: compare for example at 6.25Hz, a 26% relative WER reduction compared to a fixed-rate variant, and compared to recent work like DualCodec (Li et al., 2025) retrained for 6.25Hz improvements to 4.15% WER from 31.5% WER
  - This variable frame rate at inference (3-12.5Hz) is also shown to provide significant speedups for downstream TTS with reasonable tradeoffs in performance
- The novel ASR feature-assisted dual-stream architecture is also a strength - using features from a pre-trained ASR model, optimized for text prediction, for better semantic information than standard SSL features as validated by ablation studies in the appendix

**Weaknesses:**

- A limitation is that the decoder and downstream NAR models cannot operate directly on the variable-rate tokens: tokens must first be upsampled back to a 12.5Hz sequence via frame repetition with a (relatively) large 100M Frame Unmerging Transformer, negating some of the efficiency benefits for the synthesis stage. The Frame Unmerging Transformer, accounts for 100M of the models 216M trainable parameters
- The presented model relies on large pre-trained ASR model (a frozen 230M SenseVoice model) for the semantic features that guide the merging; other models are not compared, so it is not necessarily clear how dependent performance is on the quality and properties of this model or how generalizeable it would be to other languages or domains with weaker models

**Questions:**

- The Frame Unmerging Module seems relatively expensive for its task (upsampling repeated frames). Could a more efficient upsampling architecture (like a lightweight convolutional upsampler) potentially achieve similar acoustic quality?
- Token merging is guided by the pretrained semantic features from the pretrained ASR models. The paper's notes on L430 that acoustic and semantic information density could potentially be misaligned. Could you say more about this potential misalignment and whether for example a merging criterion based on the acoustic stream could mitigate this?
- The paper notes in the limitations the model is not streaming-capable as it operates on full sequences, but that adaptations for streaming are technically. Frame merging scans from left to right to find "maximal contiguous segments", implying a need to look ahead. How would this strategy be adapted for a low-latency streaming implementation without or only a limited look-ahead?

---

> ### Author Response · Authors · 2025-11-20
> **Response to reviewer TQuw (Part 1/2)**
>
> Dear Reviewer TQuw,
>
> Many thanks for your valuable comments and questions, which have greatly helped us improve our work. We have addressed your questions below and uploaded a revised manuscript to address your concerns.
>
> ---
>
> >  [Q1] Could a more efficient upsampling architecture (like a lightweight convolutional upsampler) potentially achieve similar acoustic quality?
>
> [A1] That is a very interesting question. Our main motivation for using a Transformer-based architecture over a convolutional one is its flexibility in handling variable upsampling rates. In our model, each token can be repeated a different number of times, **creating a highly irregular sequence for the upsampling module to process.** We hypothesize that the self-attention mechanism in the Transformer is better suited to dynamically adapt to this irregularity than standard transposed convolutions, since convolutions are typically designed for fixed upsampling factors.
>
> >  [Q2] Could you say more about this potential misalignment and whether for example a merging criterion based on the acoustic stream could mitigate this?
>
> [A2] Sure. The potential misalignment refers to cases where a segment of audio is acoustically complex but semantically simple (e.g., audios with background music, a pause in speech might contain background noise, or a non-speech sound like a cough or a laugh might be present., etc.) Our ASR-guided merging, which prioritizes semantics, might over-compress such segments, leading to a loss of acoustic detail.
>
> Using a separate merging criterion based on the acoustic stream (such as acoustic feature similarity) is a great suggestion. It could indeed mitigate this issue by preventing the over-merging of acoustically rich segments. However, this makes the semantic and acoustic streams misaligned in terms of frame rate, and may complicate downstream application algorithms. We think that **a hybrid approach that considers both semantic and acoustic information density could be a promising solution.** This is a valuable direction for future research.
>
> >  [Q3] How would FlexiCodec be adapted for a low-latency streaming implementation?
>
> [A3] Adapting our merging algorithm for a streaming context would indeed require a modification to handle the look-ahead. We would implement this by making FlexiCodec process speech chunk by chunk (e.g., 1 second), while implementing a fixed look-ahead buffer (e.g., a few hundred milliseconds) when processing each chunk. The merging decision for the current frame would be made based on its similarity to the frames within this buffer. This balances the latency and accuracy.
>
> >  [Weakness, 1] A relatively large 100M Frame Unmerging Transformer, negating some of the efficiency benefits for the synthesis stage. The Frame Unmerging Transformer, accounts for 100M of the models 216M trainable parameters.
>
> [A4] We agree that evaluating FlexiCodec's inference speed and consider its model size is critical for assessing the model's practical usability. Therefore, we have added **Appendix E** in our revision, that presents the efficiency evaluation of FlexiCodec. Our analysis shows that, while FlexiCodec is not the most efficient among the evaluated systems, it still demonstrates very acceptable efficiency. Its encoding and decoding RTF remains constant at 0.018 (encode) and 0.006 (decode) across different frame rate options (12.5Hz, 8.3Hz, and 6.25Hz). The low RTF values indicate that FlexiCodec only adds a very small overhead to downstream tasks like TTS, whose RTFs are usually more than 0.1. This positions FlexiCodec as a practical and scalable solution for real-world applications.
>
>
> | Model                | Encode Params | Encode RTF↓ | Decode Params | Decode RTF↓ |
> | -------------------- | ------------- | ----------- | ------------- | ----------- |
> | DAC-official-75Hz    | 22M           | 0.004       | 52M           | 0.002       |
> | Encodec-75Hz         | 7.4M          | 0.004       | 7.4M          | 0.004       |
> | SpeechTokenizer-50Hz | 68M           | 0.005       | 35M           | 0.007       |
> | WavTokenizer-75Hz    | 8.8M          | 0.005       | 62M           | 0.002       |
> | Mimi-12.5Hz          | 38M           | 0.003       | 40M           | 0.002       |
> | SemantiCodec-50Hz    | 725M          | 0.100       | 196M          | 0.650       |
> | SemantiCodec-25Hz    | 1660M         | 0.120       | 218M          | 0.643       |
> | XYTokenizer-12.5Hz   | 306M          | 0.022       | 214M          | 0.005       |
> | DualCodec-12.5Hz     | 622M          | 0.019       | 53M           | 0.002       |
> | FlexiCodec @12.5Hz    | 289M          | 0.018       | 161M          | 0.006       |
> | FlexiCodec @8.3Hz     | 289M          | 0.018       | 161M          | 0.006       |
> | FlexiCodec @6.25Hz    | 289M          | 0.018       | 161M          | 0.006       |

---

> > ### Comment · Reviewer_TQuw · 2025-11-20
> >
> > Thank you for your detailed reply! These are good and clarifying additions.
> >
> > For A1, would it be possible to add such a baseline? (Not necessarily within the rebuttal phase, but for the final version)
> > I agree with your motivation, but it would be nice to see the (likely) improvements experimentally borne out and see the potential computation/performance tradeoff.

---

> > > ### Author Response · Authors · 2025-11-21
> > >
> > > Dear reviewer TQuw,
> > >
> > > Thank you for your positive feedback and for the quick response!
> > >
> > > ---
> > >
> > > > For A1, would it be possible to add such a baseline? (Not necessarily within the rebuttal phase, but for the final version). It would be nice to see the (likely) improvements experimentally borne out and see the potential computation/performance tradeoff.
> > >
> > >
> > >
> > > Sure. We fully agree that adding a convolutional upsampler baseline would strenghten the paper and would allow us to learn more about the model's potential computation-performance tradeoff. **We commit to adding this baseline results in the final version of the paper**. We appreciate your flexibility regarding the timeline; if time permits, we will also try to complete the experiment within the rebuttal phase to provide an early look at the results. We believe this addition will strengthen the architectural analysis and offer a more complete picture for future readers.
> > >
> > > ---
> > >
> > > Thank you again for your reviewing efforts. We’re happy to address any more comments or concerns you may have.

---

> > > > ### Comment · Reviewer_TQuw · 2025-11-25
> > > >
> > > > Thank you for your engagement and willingness to incorporate our suggestions. I will keep my score (8 - good, accept).

---

> ### Author Response · Authors · 2025-11-20
> **Response to reviewer TQuw (Part 2/2)**
>
> >  [Weakness, 2] The presented model relies on large pre-trained ASR model (a frozen 230M SenseVoice model) for the semantic features that guide the merging; other models are not compared, so it is not necessarily clear how dependent performance is on the quality and properties of this model or how generalizeable it would be to other languages or domains with weaker models
>
> [A5] Thank you for your insightful comment. We agree that understanding the dependency on the semantic extractor and the model’s generalization capability is crucial. To address this, we refer to our ablation studies and conduct additional evaluations of out-of-domain and multilingual scenarios.
>
> - We analyzed the impact of the semantic model by comparing SenseVoice against other architectures in **Appendix D.1 (Table 8, rows E1 to E3)**. Our experiments compared the baseline (SenseVoice-Small last-layer features) against w2v-bert-2 (SSL features), Whisper-medium (a different ASR architecture), and SenseVoice-Small layer-averaged features (a different extraction strategy). The key findings are:
>
>
> 	1. **Necessity of** **ASR** **Semantics:** Using SSL features (w2v-bert-2, **Row E1**) significantly degrades performance (WER 27.3) compared to ASR features. Our analysis is that SSL features contain acoustic leakage (e.g., timbre) that reduces frame similarity, hindering the merging module.
>
> 	2. **Generalizability:** Replacing SenseVoice with Whisper-medium (**Row E2**) works effectively (WER 9.83), confirming our method generalizes to other semantic-rich ASR architectures.
>
> 	3. **Robustness:** Using layer-averaged features (**Row E3**) achieves performance comparable to the baseline (last-layer), demonstrating stability across different feature extraction strategies.
>
>
>
>
> - **More generalization studies**: We agree that our initial draft did not contain generalization study on other languages or out-of-domain evaluations. To address this, we have added a new section, **Appendix F,** to comprehensively evaluate FlexiCodec on out-of-domain (OOD) and **multilingual** scenarios using the Emilia dataset (spontaneous, in-the-wild speech). Our analysis is as follows:
>
>
> 	1. **English Out-of-Domain Robustness (Table 11):** We tested FlexiCodec on spontaneous English speech, which differs significantly from the audiobook training data. FlexiCodec maintains high semantic preservation and acoustic quality, performing comparably to in-domain baselines like DualCodec and XYTokenizer.
>
> 	2. **Multilingual Generalization and Adaptation (Table 12):** Despite being trained exclusively on English, FlexiCodec demonstrates strong acoustic generalization. While the semantic layer (RVQ1) naturally struggles with unseen languages, the full codec (RVQ1:8) achieves competitive WER and SIM scores on unseen languages. We then show that a fine-tuning on an unseen language, Chinese, can adapt FlexiCodec's semantic (RVQ-1) tokens and the merging scheme to the new language.
>
> ---
>
> Thank you again for your insightful and professional comments, which have helped us further strengthen and refine our work. If you have any further questions or suggestions, please let us know.

---

### Official Review · Reviewer_PDhk · 2025-11-03

**Soundness:** 3
**Presentation:** 3
**Contribution:** 2
**Rating:** 6
**Confidence:** 5

**Summary:**

This paper presents a neural audio codec / tokenizer for English-only speech. The primary novelty is:

- the use of a specialised module to dynamically alter frame rate.
- the utilization of a pretrained and frozen ASR model as the 'coarse' token of the codec, removing the need for semantic distillation.

**Strengths:**

- Writing and presentation is good.
- Overall technical novelty is incremental, but the changes introduced are worthwhile and justified.
- Results seem overall fairly good

**Weaknesses:**

- The authors justify the focus on frame-rate by arguing that alignment with the token-rate of text is important. However, this claim of importance feels anecdotal rather than well evidenced (although it is plausible).
- (minor) The choice of 'RVQ-1' as the name for the coarse/semantic token of the stream is confusing, given that it is not produced by an RVQ (rather FSQ). I understand why the authors chose this name as there is precedent elsewhere, but a better name is needed.
- Justification of the particular pretrained ASR model used for 'RVQ-1' exists, but direct comparison with previous semantic distillation methods is absent. This makes it hard to judge the impact of this change.
- The authors state that "We have not applied FSQ for acoustic quantization because FSQ is a single-layer quantization, and we have not discovered a multi-layer FSQ practice in literature.". There is a residual FSQ formulation available in a paper you already cite - "Scaling transformers for low-bitrate high-quality speech coding" by Parker et al.
- There are many comparable baselines with public checkpoints available that are not included in the reconstruction evaluation (especially in the <0.7kbps) section.
- It's good that the authors included subjective metrics (albeit dissapointingly only for downstream TTS, not for reconstruction), but the sample size is so small that the results are very weak. The conclusions drawn from these results need to be softened greatly, given that none of the differences are statistically significant.

**Questions:**

- It seems 'RVQ-rest' is trained with 24 levels, and then inferenced with 8? Are all results using the truncated RVQ? Why was it trained with more levels in this case? This needs elaboration or justification.
- The design for 'RVQ-rest' utilises RVQ, but the downstream TTS models work on the continuous embeddings from this part of the bottleneck. What is the motivation for using RVQ in this section if you're not going to use the tokens?
- How is the FSQ quantizer trained? Is it using straight-through gradient estimates?

---

> ### Author Response · Authors · 2025-11-20
> **Response to reviewer PDhk (Part 1/2)**
>
> Dear Reviewer PDhk,
>
> Many thanks for your valuable comments and questions, which have greatly helped us improve our work. We have addressed your questions below and uploaded a revised manuscript to address your concerns.
>
> ---
>
> >  [Q1] It seems 'RVQ-rest' is trained with 24 levels, and then inferenced with 8? Are all results using the truncated RVQ? Why was it trained with more levels in this case?
>
> [A1] We used RVQ1:8 (RVQ-1 to RVQ-8 layers) in the codec performance comparison sections (Section 4.1-4.4), but our results in the downstream TTS experiments (Appendix B.2) added up RVQ-1 and **all 24 RVQ-rest layers** to derive the continuous codec feature to be predicted by the NAR stage of downstream TTS. Therefore, not all results are using the truncated RVQ.
>
> We hope to explain more about our RVQ designs:
>
> - **Why we use 8 for codec acoustic evaluations**: As discussed in our experimental settings, we used 8 RVQ layers following standard practices in codec evaluation. This can ensure a fair comparison and allows other works to reuse our results easily.
>
> - **Why we train with more RVQ layers**: There are several reasons we consider:
>
>     - (1) We follow other works (Encodec [1] and DAC [2]) to **train more RVQ layers and using a truncated set of them.** This also allows for **broader downstream task applicability**. For example, Encodec and DAC are trained with 32 total RVQ layers. Although their downstream tasks such as VALL-E and MusicGen only use 8 of them, in other applications, such as audio compression and transmission, all 32 layers might be used. Encodec, DAC and FlexiCodec all applied RVQ quantizer dropout, so that users can choose to use different amount of RVQ layers based on the downstream task.
>
>     - (2) At very low frame rate, using only 8 RVQ layers is insufficient for good audio quality. 24 RVQ-rest layers provide a good upper bound audio quality of downstream tasks using FlexiCodec.
>
>
> Here is a table comparing the performance of 8 RVQ layers vs. all 24 RVQ-rest layers:
>
> |                            | Bitrate (kbps) | SIM  | PESQ |
> | -------------------------- | -------------- | ---- | ---- |
> | GroundTruth                | -              | 1.0  | 4.5  |
> | FlexiCodec @ 6.25Hz, RVQ1:8  | 0.64           | 0.71 | 2.76 |
> | FlexiCodec @ 6.25Hz, RVQ1:25 | 1.91           | 0.86 | 3.33 |
>
> >  [Q2] The downstream TTS models work on the continuous embeddings from this part of the bottleneck. What is the motivation for using RVQ in this section if you're not going to use the tokens?
>
> [A2] Thanks for the insightful question. Our motivations are as follows:
>
> - **The continuous NAR stage** _**implicitly**_ **predicts RVQ-rest** _**tokens**_**.** Our 12.5Hz NAR-stage TTS predicts the sum of all RVQ layer embeddings. We consider this equivalent to predicting all 1+24 layers of RVQ tokens in one pass.
> - **Continuous NAR stage allows FlexiCodec-TTS to compete with SOTA TTS solutions.** We have considered developing a VALL-E or SoundStorm-like NAR stage. However, this scheme is expected to give worse audio quality than our current solution: If we predict 8 FlexiCodec RVQ-rest layers, its audio quality upper bound is low. On the other hand, predicting all 24 RVQ-rest layers have good upper bound but likely requires a large model to predict them accuractely (because there are too many prediction heads).
>
>
>
> >  [Q3] How is the FSQ quantizer trained? Is it using straight-through gradient estimates?
>
> [A3] Yes, it uses straight-through gradient estimates. We have added this information to the revised manuscript. Thanks for pointing it out!
>
> >  [Weakness,1] The authors justify the focus on frame-rate by arguing that alignment with the token-rate of text is important. However, this claim of importance feels anecdotal rather than well evidenced
>
> [A4] We find a direct evidence in the paper "Scaling speech-text pre-training with synthetic interleaved data" by Zeng et al [3]. As illustrated in its Figure 3(a), the authors found that decreasing the audio token rate from 50Hz to 6.25Hz consistently improves the speech language model's prediction accuracy. Although the authors ultimately chose a 12.5Hz tokenizer for a trade-off between semantic preservation and frame rate, the work showed that a lower frame rate is worth exploring for improving speech language models. We have added this reference to our revised manuscript in **Line 53**.

---

> > ### Author Response · Authors · 2025-11-20
> > **Response to reviewer PDhk (Part 2/2)**
> >
> > >  [Weakness,5] An available residual FSQ implementation in TAAE.
> >
> > >  [Weakness,6] Missing low-bitrate baselines, especially under 0.7kbps.
> >
> > >  [Weakness,7] The conclusions drawn from subjective TTS metrics need to be softened since the sample pool is small.
> >
> > [A5] We have revised the footnote in **Page 6** to reflect the available residual FSQ implementation, and that we plan to investigate applying it in future work. In **Table 5**, we have added comparison with TAAE [4] (0.4kbps and 0.7kbps) and SemantiCodec[5] (0.3kbps and 0.6kbps). We have also revised Appendix B.2 to make sure MOS differences that are not statistically significant are not used to draw conclusions.
> >
> > ---
> >
> > Thank you again for your insightful and professional comment, which made our work more complete and solid! If there're any further questions, please let us know.
> >
> >
> >
> > References:
> >
> > [1] Défossez, A., Copet, J., Synnaeve, G. and Adi, Y., 2022. High fidelity neural audio compression. _arXiv_ _preprint_ _arXiv:2210.13438_.
> >
> > [2] Kumar, R., Seetharaman, P., Luebs, A., Kumar, I. and Kumar, K., 2023. High-fidelity audio compression with improved rvqgan. _Advances in Neural Information Processing Systems_, _36_, pp.27980-27993.
> >
> > [3] Zeng, A., Du, Z., Liu, M., Zhang, L., Jiang, S., Dong, Y. and Tang, J., 2024. Scaling speech-text pre-training with synthetic interleaved data. _arXiv_ _preprint_ _arXiv:2411.17607_.
> >
> > [4] Parker, J.D., Smirnov, A., Pons, J., Carr, C.J., Zukowski, Z., Evans, Z. and Liu, X., 2024. Scaling transformers for low-bitrate high-quality speech coding. _arXiv_ _preprint_ _arXiv:2411.19842_.
> >
> > [5] Liu, H., Xu, X., Yuan, Y., Wu, M., Wang, W. and Plumbley, M.D., 2024. Semanticodec: An ultra low bitrate semantic audio codec for general sound. _IEEE Journal of Selected Topics in Signal Processing_.

---

> ### Comment · Reviewer_PDhk · 2025-11-24
> **Productive review process**
>
> Thanks to the authors for a very productive review process. They have addressed my concerns (and those of the other reviewers), and I feel the paper is significantly improved. I'm happy to raise my score to an 8.

---

> > ### Author Response · Authors · 2025-11-25
> > **Thanks to reviewer PDhk**
> >
> > Dear reviewer PDhk,
> >
> > Thank you for taking the time to review our rebuttal. We are glad to have addressed your concerns! Once again, we appreciate your detailed and helpful review, as well as your positive feedback.

---

### Author Response · Authors · 2025-11-20
**General response to all reviewers**

Dear Reviewers,

We sincerely thank you for your detailed, insightful, and constructive feedback. **We have now uploaded a revised version of our paper, with the changes highlighted in red**. While we have provided responses to each reviewer individually, we wish to highlight the key improvements and additions made in this revision for your convenience:

- Efficiency Analysis (**Appendix E**): To assess the model's practical usability, we added a new section benchmarking the Real-Time Factor (RTF) and parameter counts of FlexiCodec against nine other codecs.

- Generalization Study (**Appendix F**): To address concerns about robustness and multilingual capabilities, we conducted an evaluation on the in-the-wild Emilia dataset, in contrast to the audiobook training set, assessing our codec's ability on out-of-domain English speech as well as multilingual speech.

- Factorial Ablation Study (**Appendix D.2**): To clearly display the contributions of our core design choices with respect to semantic preservation, we added a new factorial analysis. It displays the incremental impact of FlexiCodec components.

- New Baselines and In-depth Comparisons (**Table 5**): We have strengthened our main results (Table 5) by adding new, strong low-bitrate baselines TAAE and SemantiCodec. We added a discussion (**Appendix G.2**) comparing the FlexiCodec and a concurrent related work XY-Tokenizer.

- Further Discussions and Clarifications (**Abstract, Introduction, Related Works, Footnotes, Appendix J**): We have revised the manuscript to make our motivation more clear and acknowledges the semantic-acoustic tradeoffs. We added footnote on the bitrate calculation for frame lengths, residual FSQ, and expanding the Future Work (**Appendix J**) section with discussions on learnable merging algorithms and connections to syllabic units.


Thank you again for your valuable input! We are available for any further questions or discussion.

---

### Author Response · Authors · 2025-12-04
**Thank You to the ICLR Reviewing Committee**

We sincerely thank all six reviewers for their diligent efforts in reviewing our submission. We are especially grateful to the Area Chairs (ACs) and Senior Area Chairs (SACs) for dedicating additional time and effort to this year's ICLR review process.

During the rebuttal period, we were fortunate to **receive responses from 3 reviewers, 2 of whom raised their scores** before they were reverted due to the incident. All 3 responding reviewers acknowledged that our revisions and responses had addressed their questions and improved the paper. Here is a summary of their replies and score changes:
- **Reviewer PDhk** increased their overall rating from 6 (Confidence 5) to 8 (Confidence 5), commenting that the review process was productive and that the revisions significantly improved the paper.
- **Reviewer TQuw** maintained their overall rating of 8, acknowledging that our additions were good and clarifying.
- **Reviewer U5qj** increased their overall rating from 8 to 10 and raised their "Presentation" and "Contribution" ratings from 3 ("Good") to 4 ("Excellent").

We once again thank all the reviewers for their valuable suggestions, which have helped us improve our work. We also extend our gratitude to the Area Chairs, Senior Area Chairs, and Program Chairs for your hard work during this challenging review cycle.

---

### Meta-Review · Area_Chair_oWEG · 2026-01-06

**Summary:**

- A shared concern across several reviewers was whether the reliance on pretrained ASR features would inhibit its effectiveness in OOD language. However the authors effectively addressed this concern with additional multi-lingual experiments.
- The authors also adequately addressed additional questions regarding real time factor and factorized ablation studies for design choices.
- Reviewer URnt mentioned additional prior works so the authors further adjusted the scope of their claims to account for these additional related works.
- Overall feedback was mostly positive with several reviewers noting good results and detailed experimentation.

**Reviewer Concerns:**

I believe all technical concerns have been adequately addressed.

**Reviewer Scores:**

Based on the discussion, it's already clear that several reviewers have raised their scores to around 8 after the rebuttal. It's possible that URnt would have only raised their rating to borderline if they felt the overall impact of the work was limited given the reduced claims. However, average ratings would likely have been safely above borderline.

---

### Decision · Program_Chairs · 2026-01-26

Accept (Poster)